# Merits and Limitations of Element Balances as a Forest Planning Tool for Harvest Intensities and Sustainable Nutrient Management—A Case Study from Germany

**Bernd Ahrends** [1],*[ID]**, Klaus von Wilpert** [2]**, Wendelin Weis** [3]**, Christian Vonderach** [4]**, Gerald Kändler** [4]**, Dietmar Zirlewagen** [5]**, Carina Sucker** [2] **and Heike Puhlmann** [2]

[1] Department of Environmental Control, Northwest German Forest Research Institute (NW-FVA), D-37079 Goettingen, Germany
[2] Department of Soil and Environment, Forest Research Institute of Baden-Wuerttemberg (FVA-BW), D-79100 Freiburg, Germany; klaus.von-wilpert@online.de (K.v.W.); carina.sucker@web.de (C.S.); heike.puhlmann@forst.bwl.de (H.P.)
[3] Department of Soil and Climate, Bavarian State Institute of Forestry, D-85354 Freising, Germany; wendelin.weis@lwf.bayern.de
[4] Department of Biometry and Information Sciences, Forest Research Institute of Baden-Wuerttemberg (FVA-BW), D-79100 Freiburg, Germany; christian.vonderach@forst.bwl.de (C.V.); gerald.kaendler@forst.bwl.de (G.K.)
[5] INTERRA, Bureau for Environmental Monitoring, St. Peter Str. 30, D-79341 Kenzingen, Germany; info@interra.biz
* Correspondence: bernd.ahrends@nw-fva.de

**Abstract:** Climate change and rising energy costs have led to increasing interest in the use of tree harvest residues as feedstock for bioenergy in recent years. With an increasing use of wood biomass and harvest residues, essential nutrient elements are removed from the forest ecosystems. Hence, nutrient sustainable management is mandatory for planning of intensive forest use. We used soil nutrient balances to identify regions in Germany where the output of base cations by leaching and biomass utilization was not balanced by the input via weathering and atmospheric deposition. The effects of conventional stem harvesting, stem harvesting without bark, and whole-tree harvesting on Ca, Mg and K balances were studied. The nutrient balances were calculated using regular forest monitoring data supplemented by additional data from scientific projects. Effective mitigation management strategies and options are discussed and calculations for the compensation of the potential depletion of nutrients in the soil are presented.

**Keywords:** soil nutrient balance; deposition; weathering; leaching; uncertainties; harvest intensities; forest monitoring data; Germany; National Forest Inventory (NFI); National Forest Soil Inventory (NFSI)

## 1. Introduction

The supply of base cations like sodium (Na), potassium (K), magnesium (Mg) and calcium (Ca) into forest soils occurs through the dissolution of minerals and inputs through atmospheric deposition [1]. The decline in base cation deposition throughout large parts of Europe [2,3] will partly offset the positive effect of reduced base cation leaching due to decreased sulphur deposition [4,5], even if the sulphur emissions in Germany have decreased by more than 90% in the past decades [6]. Additionally, forests are currently and will be in the coming decades under pressure to fulfil the rising demands for timber and biomass as a sustainable energy source [7], which can contribute to reducing greenhouse gas emissions [8,9].

With the intensifying use of wood biomass and harvest residues, like tree tops, branches and bark, the associated nutrient element exports increase disproportionately [10], as the element concentrations in these tree parts are much higher than in stem wood [11,12].

Subsequently, the additional exports of base cations may have significant impacts on soil element stocks and soil quality [13], if weathering or other input fluxes cannot compensate for the resulting losses. Under these conditions, a recovery of forest soils from past acidification is not to be expected, despite a considerable ongoing decrease of sulphur deposition in Europe [4].

Accordingly, concerns have been raised about the sustainability of harvesting practices and their net impact on forest productivity, particularly during the second and subsequent rotation periods [14]. Therefore, a world-wide debate on the merits and trade-offs of additional forest biomass use is under way in the scientific community, as well as in politics, forestry practice and among certification authorities. Practical decisions in harvest intensity planning have long-term consequences on soil quality, forest growth and potentially necessary compensation measures. However, decisions are often based on intuition [15] or very different, hardly comparable methodological approaches [16]. As indicators should have a scientific basis and be applied operationally in a mapped way [17], a common method for site-specific assessment is to calculate input-output nutrient budgets [7,18,19].

In the Netherlands, soil nutrient balances were calculated in order to develop nation-wide forest harvesting guidelines specific to regions of comparable deposition, tree species and soil types [7]. For each region-tree-soil combination, the maximum possible harvest intensities were calculated under equalized nutrient balance including uncertainties. The results show that on poorer sandy soils, even the current rates of timber and biomass exports lead to negative balances, particularly for phosphorus (P) and K. A long term mass balance study in Sweden showed net losses of Ca and Mg for stem harvesting and whole-tree harvesting (WTH) scenarios throughout most parts of the country [20]. In another study from Sweden, WTH reduced the base saturation in the soil [21]. Long-term base cation balances for forest soils in Finland demonstrated that WTH will lead to the depletion of base cations [22]. In 1066 Finish lake catchments, stem-only and stem-and-branches harvesting scenarios resulted in a balanced base cation budget, whilst WTH scenarios depleted the soil base cation pools [13]. In contrast, the calculations of Forsius et al. [23] for Finnish forests suggested that the input by weathering and deposition was sufficient to sustain the nutrient demand of WTH. For the British Isles, the predicted input from weathering and deposition were sufficient to compensate the losses of Ca, Mg, and K for stem-only and stem plus branch harvest scenarios [24]. Whole-tree harvesting resulted in a negative Ca balance at about half of the studied sites. Consistent with this, one of the oldest European WTH experiments in the United Kingdom also observed a significant decrease of soil base saturation after harvesting all above ground biomass [14]. When comparing these studies from different countries, it is important to note that conventional harvest schemes vary widely across countries [25] and that the considered WTH scenarios differ as well (for example in terms of the harvested tree compartments).

A review by Agate et al. [26] confirmed that high nutrient losses from soils have measureable consequences for forest ecosystems. Most studies therein revealed a negative effect on stand growth (tree diameter, tree height and tree volume) with a 3–7% reduction in the short and medium-term (up to 33 years after use), especially when canopy biomass including foliage was exported. The review by Thiffault et al. [27] also indicated medium-term ($\leq$24 years) growth reductions in intensively used stands. However, the long-term effects are largely unknown [28] and it should be noted that the negative effects on stand growth after WTH observed in some studies resulted from a temporary nitrogen deficiency [29]. The high risk of site degradation due to intensified biomass harvesting [30] makes it necessary to develop forest management strategies which will not impair forest productivity in the long term [8] and which take into account that the risks are highly dependent on soil fertility, stand and site conditions [31,32].

In Germany and many other countries, there is a great interest in sustainable nutrient management in the forestry sector. Nutrient balances are widely adopted tools to assess sustainable forestry. However, the calculation procedures in the individual federal states of Germany differ significantly from each other and the approaches have only been imple-

mented in small-scale studies. For example, very different models and methods are used to calculate weathering [33–36] and leaching rates [19,37,38].

The objective of our study was to improve the nationwide assessment of the effects of harvest intensities on sustainability of the element budgets of forest soils. Specifically, we aimed at: (1) calculating methodologically uniform nutrient balances and their uncertainties for Germany using established, regular environmental monitoring systems as spatial data basis; (2) analysing the spatial patterns of harvesting effects on element balances to assess potential and risk of actual and intensified biomass harvesting for German forests and (3) deriving and quantifying strategic approaches for adapting harvest intensities as well as other nutrient management options to the actual nutrient availability at a regional scale.

## 2. Materials and Methods

### 2.1. Calculation of Nutrient Balances

Soil nutrient balances are commonly used indicators in both agriculture and forestry where annually or periodically aggregated balances express changes in soil nutrient stocks and soil fertility. Compared to agricultural systems, where fertilizer input dominates the soil nutrient balance, calculating soil nutrient balances for forest ecosystems poses a much greater challenge and the involved uncertainties are very high [39]. In forest soils, the relevant processes are inputs by atmospheric deposition as well as soil mineral weathering and the output fluxes by leaching and forest harvesting (Figure 1). For the assessment of different management options, the calculations should be performed in two steps: (1) calculation of nutrient balances without harvest removal (environmental part) and (2) calculation of total nutrient balances considering specific management options, e.g., different harvest scenarios or soil protective liming.

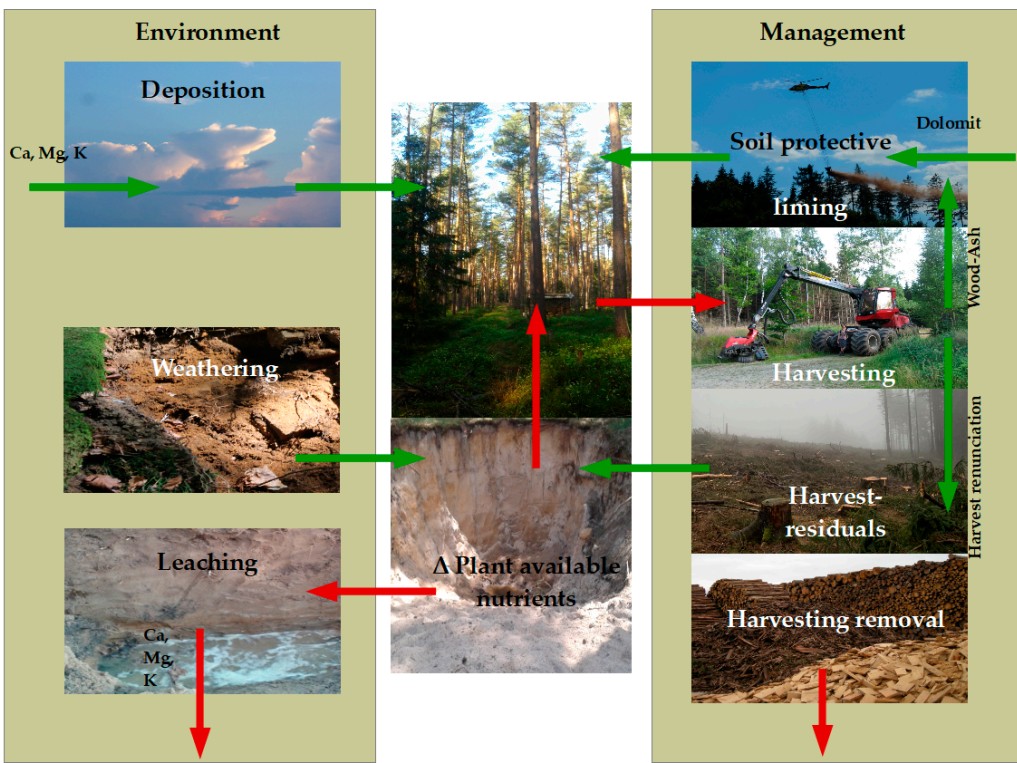

**Figure 1.** Schematic view of considered element fluxes for soil nutrient balances of Ca, Mg, and K in forest ecosystems and their relationships to each other. One 1000th of the plant-available nutrient stock in the soil is taken into account as an additional input (buffer).

In our study, the calculation of nutrient balances was restricted to the main nutrients Ca, Mg and K. These elements largely determine the resilience of forest soils to acidification.

Nitrogen (N) fluxes were not calculated because high nitrogen emissions in the recent past and at present have caused an oversupply of nitrogen throughout Germany [19,40]. Phosphorus, also an essential nutrient for tree growth and health, is characterized by very small fluxes in both leaching and deposition, and P concentrations are often near or below the limit of detection of conventional analytical methods [41]. In addition, part of the P transport in soils takes place in colloidal inorganic and organic P forms and can only be measured with considerable analytical effort [42]. Phosphorus was therefore not addressed in this study.

In some forest soils, the depletion rate of base cations in relation to element pools is very small [24]. To take this into account, one thousandth of the plant-available soil nutrient stock of Ca, Mg and K up to a depth of 90 cm was added to the nutrient balance and thus would be depleted in 1000 years at the earliest. Such a buffer is appropriate because small balance deficits are more tolerable in soils with high plant-available nutrient stocks and natural soil acidification is an extremely slow process [43]. All presented balances were averaged over the years 2000 to 2010 (data and conditions of the second National Forest Soil Inventory (NFSI II)) to compensate for annual fluctuations.

### 2.1.1. Study Sites

Around one third (32%) of Germany is covered by forest. Due to the high diversity of soils in Germany, reflecting different landscapes formed during the ice age, climate, and bedrock conditions, we stratified Germany into eight different model regions or soil landscapes (Figure 2A).

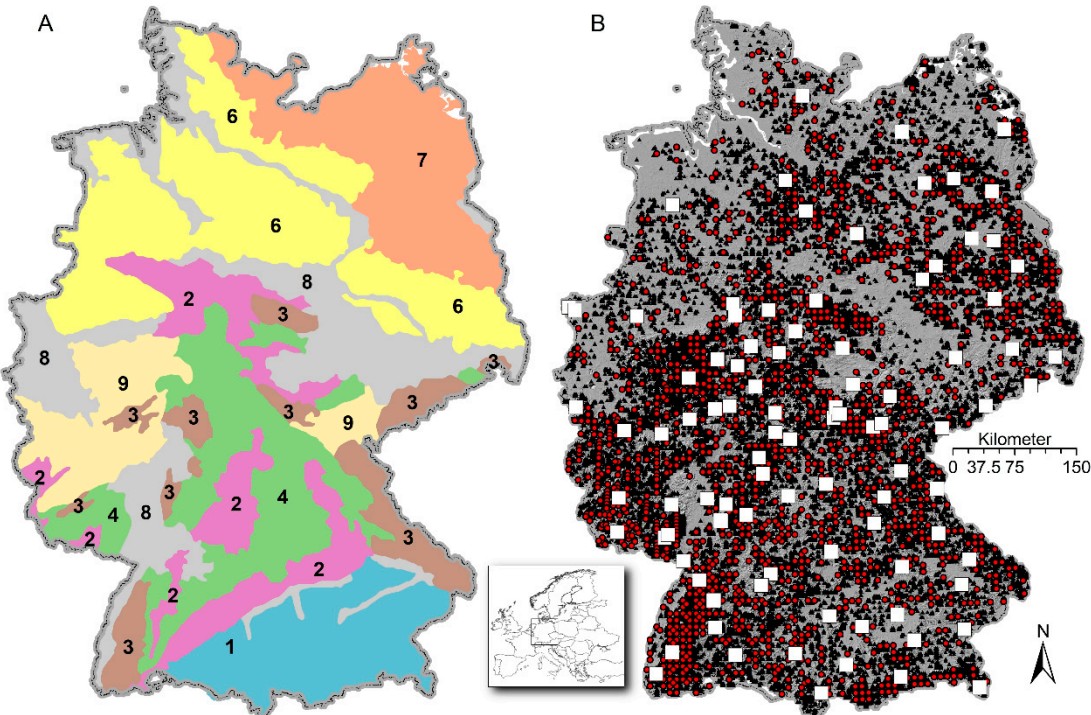

**Figure 2.** Stratification of Germany in soil landscapes (model regions—(**A**)) and location of intensive monitoring plots (□), National Forest Soil Inventories (NFSI) plots (●) and National Forest Inventory (NFI) tracts (▲) in Germany (**B**). Legend of left map: 1—pre-alpine moraines and limestone Alps; 2—hills on limestone bedrocks; 3—hills on crystalline bedrocks; 4—hills on sand, silt, clay bedrocks; 6—old moraines, north German lowland; 7—young moraines, north German lowland; 8—loess regions, fluvial valleys; 9—hills on clay- and silt schist bedrocks. Note: number 5 was deliberately not assigned.

For model development, application, regionalization, and evaluation, we used data from three different monitoring networks in Germany (Figure 2B). Data from Intensive

Forest Monitoring sites were used to parametrize the statistical deposition model (cf. Section 2.1.3) and for the estimation of weathering rates (cf. Section 2.1.4) and element concentrations in seepage water (cf. Section 2.1.5). Data from an 8 × 8 km grid of the NFSI II (1690 sites) were used to calculate weathering and leaching rates and then transferred from these sites to the 23,880 tracts of the third German National Forest Inventory (NFI 2012), using statistical regionalization methods (cf. Section 2.2). The German NFI is based on a systematic rectangular grid with clusters (tracts) as primary sampling units. The sample grid has a width of 4 × 4 km covering the entire forest area of Germany. The NFI data can be used to quantify different forest use scenarios and compare these to the nutrient balance without harvesting (WOH). In addition, the 16-fold higher data density of the NFI compared to the NFSI allows a more spatially differentiated identification of regions with critical nutrient balances.

### 2.1.2. Model Formulation

The total nutrient balance (EB) with harvest (WH) was calculated according to:

$$EB_x = DEP_x + WEA_x + LEA_x + HAR_x + \frac{S_x}{1000} \qquad (1)$$

for each element x (Ca, Mg, K) separately, where DEP is the deposition, WEA the weathering rate, LEA the leaching, HAR the harvest removal of base cations and S denotes one 1000th of the plant-available nutrient stock in the soil. In scenarios without harvest (WOH), HAR was set to zero. All fluxes were calculated as kg ha$^{-1}$ yr$^{-1}$ and for the comparison between different charged ions, kmol$_c$ ha$^{-1}$ yr$^{-1}$. Following Sverdrup et al. [44], cation exchange and the release of base cations in the decomposition process of organic matter were not taken into account, because they are internal cycles and accordingly not long-term sources.

We defined three different potentially feasible harvesting scenarios along a gradient of harvesting intensity (MIN, REAL, MAX—see Table A1). The scenarios considered different intensities of stem and crown utilization as well as the redistribution of biomass in the forest stand, its accumulation on skid trails or its export depending on the utilization technique. Scenario MIN was the most resource saving harvest intensity where only saw logs and industrial wood with diameters > 12–17 cm are harvested. Scenario REAL represented a common harvest intensity where saw logs, industrial wood, and fuel wood with diameters > 7 cm are used. In the most intense scenario MAX, all above-ground woody biomass except inevitable harvesting losses is used. More details can be found in Table A1.

### 2.1.3. Deposition

The estimation of total deposition (TD—sum of wet, dry and occult) of the base cations x (Ca, Mg, K) was based on combined regionalized measurements (wet deposition [45,46]) and canopy budget model calculations (dry and occult) after the 'filtering approach' [47] for the period 2000–2010. The calculations are performed as follows:

$$TD_x = BD_x + BD_x \cdot DDF \qquad (2)$$

and the dry deposition factor (DDF) is calculated from Na deposition [48,49]:

$$DDF = \frac{(TF - BD)_{Na}}{BD_{Na}} \qquad (3)$$

where TF = throughfall deposition and BD = bulk deposition. This approach assumes that Ca, Mg and K aerosols are deposited with equal deposition velocity as Na particles.

Generalized additive mixed models (GAMM) [50] were used to explore and quantify the impact of forest stand characteristics, topographic and atmospheric variables on DDF. For this, measured deposition data [51] from more than a hundred Intensive Forest Monitoring sites with a variety of forest stand types were analysed. The final DDF model

included the predictor variables wind speed, windward and leeward effects, distance to the North Sea (as proxy for sea salt concentration), bulk deposition, tree species and stand height. Bulk open field deposition was estimated from regionalized wet deposition maps which were adjusted using correction factors for Germany from Gauger et al. [52]. The complete model with all parameters and validation results is summarized in Appendix B. The calculation of the GAMM models was performed with R 3.01 software [53], package "mgvc", landscape morphology was analysed with SAGA [54].

### 2.1.4. Weathering Rates

Weathering rates in the mineral soil were calculated with the geochemical model PRO-FILE [55], which has been frequently applied in Europe [56–58] and North America [59–61]. The particularly sensitive input variables [62,63] were parameterized as follows: The specific surface area (SSA) was calculated using the equation from Phelan et al. [60], which is a modification of the equation from the original PROFILE model [55] and is also valid for soils with clay contents of more than 20%. Dynamic soil water contents for all NFSI profiles were derived from water budget modelling using LWF-Brook90 [64]. A detailed description of the water budget model parameterization is given in a variety of recent studies [65–67]. The mean annual soil temperature was taken from regionalized climate data (cf. Section 2.1.5). As mineral analyses were not available for most of the NFSI sites, the mineralogical input to PROFILE was estimated from total geochemical soil analyses with the A2M ("Analysis to Mineralogy") model [68].

### 2.1.5. Leaching of Base Cations

Leaching rates were estimated by multiplying the amount of annual seepage water with an estimated element concentration in the seepage. Plot-specific soil water fluxes were estimated with the physically based hydrological model LWF-Brook90. The LWF-Brook90 model requires meteorological input data in daily resolution (precipitation, temperature, radiation, water vapour pressure, wind speed). The model was run from 2000 to 2010 using regionalized daily climate data derived from measurements at the weather stations of the German Weather Service (German: Deutscher Wetterdienst, DWD). Temperature, vapour pressure and wind speed were interpolated using GAMs, precipitation, and global radiation by kriging. Methodical details and information on model performance are given by Ahrends et al. [65].

To address the second methodological challenge—estimating element concentrations in seepage water—data on soil water extracts measured on the NFSI plots were used. An example for the estimation of seepage nitrate concentrations from water extracts measurements is described in Fleck et al. [19]. Sample preparation and analysis for the soil water extracts followed standardized procedures according to the guidelines for harmonized methodologies for laboratory analyses [69–72].

The exact procedure is described in Weis et al. [73] and includes the following main steps: (1) estimation of seepage concentrations of strong anions (sulphate, nitrate, chloride) from their concentrations in the aqueous soil extracts (water to soil ratio 2 ÷ 1); (2) estimation of inorganic dissolved carbon concentration from soil pH in water; (3) estimation of the molar fractions of the cations in the leachate from their fractions of the effective cation exchange capacity (extraction with 1 M $NH_4Cl$ solution; for carbonate-containing soils extraction with 0.1 M $BaCl_2$ solution); (4) multiplication of the estimated cation fractions with the total anion concentration.

The approach was based on the following simplifying assumptions: the base cation leaching is mainly driven by the leaching of the anions sulphate ($SO_4^{2+}$), nitrate ($NO_3^-$) [7] and chloride ($Cl^-$) and the anion discharge is equal to the cation discharge; organic anions can be neglected in the discharge horizon; the cation fractions in the seepage water can be predicted with sufficient accuracy from the cation fractions at the soil exchanger; the total concentration (activity) of the ions in the seepage water plays a subordinate role. At 90% of the NFSI sites the discharge horizon was assumed to be in the mineral soil at depth of

60–90 cm; at 10% of the sites the soil development was shallower, and thus the surface of the bedrock was taken as discharge horizon there.

2.1.6. Estimation of Nutrient Export under Different Harvest Intensities

The National Forest Inventory in Germany is the primary source of national forest information and has been conducted three times so far (1987, 2002 and 2012). To project forest development and timber supply based on NFI data into the future, the empirical single-tree growth model WEHAM [74,75] was used. For this study, the resulting data were used to define initial conditions for the simulation of harvest scenarios. Normally the NFI is not to be evaluated for individual tracts, but tracts are aggregated into larger areas comprising all age classes of relevant forest types. When calculating nutrient balances for each NFI tract, we had to assume that nutrient removal by harvest corresponded to the long-term average. Hence, we used real removals during 2002 to 2012 (based on NFI data) and WEHAM projections for 2012 to 2052 encompassing 50 years of forest development. Evaluation results showed that calculated nutrient balances from the nutrient removal at the NFI tracts were independent of stand age (Figure A3).

For our nutrient balances, the exported biomass for each NFI sample tract and each scenario (Table A1) was calculated based on the available biometric tree information using additive biomass functions for different species and different components: stump, stump bark, solid wood, bark of solid wood, brushwood with a diameter of less than 7 cm and needles (if applicable; leaves for broadleaved trees were not considered) [76]. Taper curves and assortment algorithms were used including the quantification of bark share [77]. Mean contents of Ca, Mg and K in the biomass compartments "coarse wood", "bark of coarse wood" and crown biomass (brushwood plus bark and needles when coniferous) were calculated from twelve scientific studies [33,78–88] and summarized in Rumpf et al. [12]—including 451 experimental sites and 1498 trees of 5 coniferous and 6 broad-leaved tree species. The element exports with harvest were derived from the multiplication of biomass values and nutrient contents of the respective tree components based on GAM models. The pre-defined utilization scenarios (Table A1) determined the average nutrient export at each NFI sample tract.

*2.2. Regionalization of Nutrient Fluxes and Soil Stocks*

The data needed for calculating the soil-based components of the nutrient balances as well as the balances themselves were only available (measured or assessed by transfer functions) at the grid points of the NFSI. None of these data were available at the systematic grid with tracts of the National Forest Inventory (NFI). Therefore, results were transferred from the NFSI grid to the NFI tracts by means of regression models (stepwise multiple regression models combined with kriging of model residuals when needed) on log transformed response variables (alternatively tested boosted regression trees were outperformed according to the model validation). As predictors, we used quasi-continuously available key variables like geology, topography, soil types, climate and deposition from nationwide data bases and maps (GÜK2000 for geological overview, BÜK50 or BÜK200 for soil types, a 25 m DEM grid for deriving topographical indices and deposition maps [46]). Moreover, information on forest conditions (proportion of coniferous trees, crown condition, soil protective liming) were included in the regression models as potential co-variables.

The transfer was performed in three steps: First, the data was split in half resulting in a training and a validation data set, both randomly distributed in space. Second, the regression models were parameterized (ordinary least squares, OLS) separately for each soil landscape (cf. Figure 2A) at the NFSI grid points. An assessment of the extent to which the regionalized balances reflected the distributional characteristics of the balances based on the measured NFSI data is provided in Figure A6. Model performance was tested after back-transformation and bias correction [89] in terms of $R^2$ and RMSE (Root Mean Square Error) using the validation data set. Finally, the regression models were applied with their respective individual set of predictors to each of the NFI tracts with available metadata

except for deposition, which was estimated directly for all NFI tracts (cf. Section 2.1.3 and Figure A7).

The completeness of measured parameters at the sampling points of the NFSI differs between the federal states, even if a big effort has been invested for harmonizing the environmental monitoring systems for all of Germany [72,90]. Data availability differed between the balance components: data for calculating weathering rates were available at 86% of the sites, for nutrient leaching with seepage water at 66% and for nutrient balances at 55%. Hence, only at around half of the NFSI points, could the complete nutrient balance including all balance components be calculated. Therefore, transfer of the nutrient balance from the NFSI points to the NFI tracts was achieved in two steps: (1) transfer of the individual balance terms from the NFSI points to the NFI tracts, (2) calculating the nutrient balance at the NFI tracts from the transferred balance components.

For the stratified model regions (Figure 2A), different regionalization models were parameterized. The regions 6 and 7 were combined for the final model selection as both are characterized by glacial till and comparable maritime climate.

Additionally, global models for whole Germany were parametrized for comparison of the model performance with the stratified models. The $R^2$ for the Ca, Mg and K balances were by 0.28, 0.29 and 0.18 lower in the global models than in the stratified ones. An overview on the indicators of model performance of the stratified models is given in Table 1. The model errors were highest for Ca and Mg in model regions with limestone bedrocks (region 1 and 2, Figure 2A) and amounted to a multiple of the mean balance level, whereas for K, model errors were comparatively small and quite uniform for all model regions with a $CV_{RSME}$ (standardized RMSE) of around 0.5. All regression analyses were performed with R environment for statistical computing and the following software packages: "OLS", "randomFOREST", "stats" and "dismo". Landscape morphological indices were calculated with SAGA [54].

**Table 1.** Indicators of model performance for the soil-related balances (weathering + deposition—leaching with seepage) in the validation data set. OBS: number of observations; $R^2$: coefficient of determination; RMSE: root mean squared error [kg ha$^{-1}$ yr$^{-1}$]; $CV_{RMSE}$: standardized RMSE.

| Region | OBS | Ca Balance | | | Mg Balance | | | K Balance | | |
|---|---|---|---|---|---|---|---|---|---|---|
| | | $R^2$ | RMSE | $CV_{RMSE}$ | $R^2$ | RMSE | $CV_{RMSE}$ | $R^2$ | RMSE | $CV_{RMSE}$ |
| 1 | 63 | 0.733 | 118.580 | −8.024 | 0.640 | 35.193 | −5.940 | 0.665 | 3.990 | 0.454 |
| 2 | 44 | 0.679 | 111.700 | 2.877 | 0.015 | 38.815 | 2.870 | 0.509 | 4.726 | 0.434 |
| 3 | 79 | 0.690 | 10.000 | −2.421 | 0.527 | 6.809 | −2.625 | 0.523 | 4.125 | 0.563 |
| 4 | 126 | 0.540 | 25.631 | 4.626 | 0.226 | 8.932 | 3.369 | 0.344 | 5.213 | 0.465 |
| 6\|7 | 46 | 0.650 | 7.775 | 4.527 | 0.587 | 1.992 | 0.704 | 0.800 | 1.546 | 0.305 |
| 8 | 31 | 0.584 | 36.587 | −14.405 | 0.339 | 4.168 | 1.447 | 0.382 | 4.604 | 0.611 |
| 9 | 61 | 0.623 | 12.438 | −1.553 | 0.387 | 6.041 | −1.841 | 0.475 | 3.736 | 0.485 |
| Global | 450 | 0.605 | 62.514 | −2.054 | 0.447 | 23.164 | −0.288 | 0.521 | 4.313 | 0.474 |

### 2.3. Treatment of Special Sites

On sites dominated by limestone and dolomite, the supply of Ca and Mg to forest soils and trees is usually unlimited. The high solubility of these carbonates causes large Ca and Mg fluxes from weathering and in seepage flux. In combination with equally high uncertainties, this discourages a sufficiently reliable interpretation of Ca and Mg balances for these sites. Therefore, the element balances for Ca and Mg were assumed to be even on limestone and dolomite. The occurrence of carbonate was predicted with a logistic classification model (recall accuracy >90%), which was calculated using the R packages "logistf", "stats" and "dismo".

The Ca and Mg balances are also influenced by liming, as both elements are contained in the applied dolomitic limestone. The dissolution of dolomites initially increases the input of both elements into the soil. As a result, higher leaching exports may occur temporarily,

but also in the longer term [91,92]. Accordingly, liming effects should be considered in the balances of Ca and Mg. Information on past liming events is available for NFSI plots, although it is very heterogeneous with respect to timing, repetition, and amount of dolomite application in the different federal states of Germany. For the NFI tracts, reliable data on liming are missing for parts of the federal territory or are distributed very heterogeneously in some states. However, evaluations within the framework of the regionalization of input and output fluxes showed that, where liming was documented, the liming effect in the regionalization models was only rarely significant with respect to the target variables leaching, weathering and soil stocks for Ca and Mg and had very low sensitivity on model results with changing signs [93]. This shows that liming obviously has ambiguous effects in our data and, at best, only very weak influence on the nutrient balances of Ca, Mg and K up to a depth of 90 cm. Accordingly, the liming effect was also not considered in the German-wide calculations of this study.

### 2.4. Uncertainty Estimations

Usually, calculating soil nutrient balances is associated with a high degree of uncertainty, mainly due sampling and measurement errors, errors of the predictive models for balance elements as well as regionalization errors and biases [44,94,95]. Although uncertainty estimation is an important part of model application, there are numerous challenges and pitfalls. A thorough, very detailed, statistical discussion of uncertainty may reduce acceptance by stakeholders [96] as would the concealment of prediction uncertainties [97]. Yanai et al. [98] demanded that element balances should be supplemented with uncertainty analyses as a standard tool, not least also to allow reliable statements about the significance of any presented results.

The Monte Carlo simulation method is a widely used technique for uncertainty analysis, which can be described as follows [99]: For a model $\Phi$ of arbitrary complexity, the calculation of the resulting variable Z is done according to:

$$Z_i = \Phi(X_i, Y_i) \tag{4}$$

where X and Y are normally distributed random variables and the index i refers to samples from these normal distributions. X and Y are assumed to be independent of each other and covariance terms are not taken into account directly [63]. However, Yanai et al. [98] recommended to consider each covariance structure in their joint probability distributions when randomizing the parameters. In our application, the variation of the parameters was generated with the function rmvnorm() from the R software package "splus2R" [100] directly incorporating the covariance between the variables. A compilation of statistical parameters used for the different parts of the element budget calculations and to account for regionalization errors can be found in Appendix C.

We performed repeated calculations of the nutrient balances according to the respective balance equation. An error value randomly selected from its known (or assumed) probability distribution was repeatedly added to the model prediction for the individual balance elements, or the sub-equations for their calculation (e.g., leaching, harvest removal). After 10,000 iterations, the magnitude of the total error was derived from the realized predictions and the corresponding result statistics (mean, standard deviation, quantiles, etc.) were calculated. To assess the level of significance for the occurrence of negative or positive element balances, we analysed the resulting probability densities of the nutrient balances. The terms significant and weakly significant were defined with error probabilities of $\alpha \leq 0.05$ and $0.05 < \alpha \leq 0.1$, respectively.

## 3. Results and Discussion

### 3.1. Nutrient Fluxes of Deposition, Weathering, Leaching and Harvest Removal

The influence of the various balance components on the total budget calculations was very element-specific (cf. Figure 3A–C). For Ca, a relatively even distribution among the balance components was observed. Loss by leaching and removal tended to be higher than

gains from deposition and weathering, so that the overall balance is negative in many cases. Due to the high leaching losses, more than 25% of the NFI tracts had a negative Ca balance already when not considering the harvest removal. For Mg, harvest removal was less important, and the balance remained positive on average. In the case of K, leaching losses were very small and harvest removal was the dominating loss factor. At most of the NFI tracts, harvest export (scenario REAL) was compensated by weathering rate and negative balances were calculated for slightly less than 20% of the NFI tracts. The considered soil stocks (cf. Section 2.1.2) were generally of minor importance, especially for Mg and K. To give an idea of the magnitude and large-scale spatial differences of the balances and their input variables, the medians for different model regions (see Figure 2A) are compiled in Table 2.

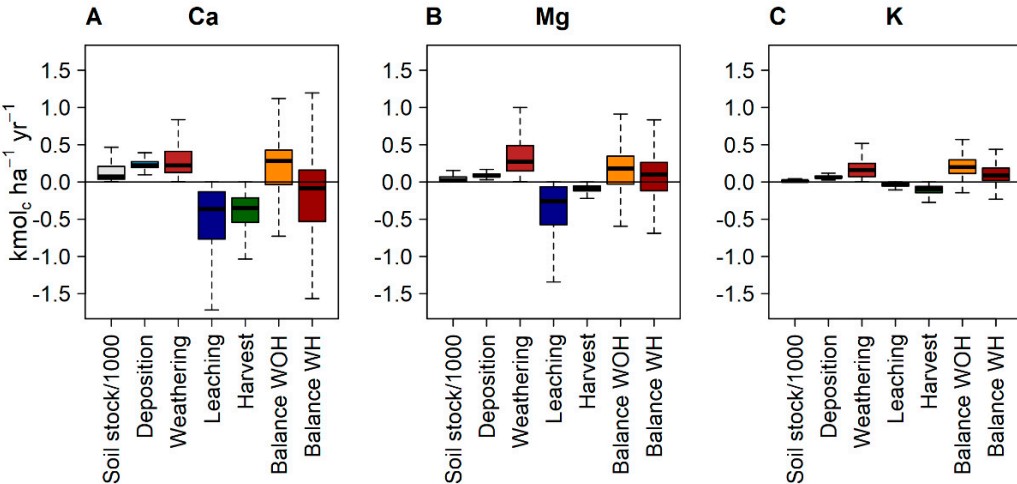

**Figure 3.** Median and range data for calcium (**A**), magnesium (**B**) and potassium (**C**) of soil stocks, deposition, weathering, leaching, harvest removal, nutrient balance without harvest (WOH), and nutrient balance with harvest (WH) for NFI tracts in Germany (for Ca and Mg excluding carbonate sites; harvest export for scenario REAL).

The weathering rates of the base cations (Ca, Mg, K) in the stratified model regions (see Figure 2A and Table 2) were, in ascending order, as follows: young moraines, north German lowland: 0.3, old moraines, north German lowland: 0.35, hills on clay- and silt schist bedrocks: 0.7, hills on crystalline bedrocks: 0.7, hills on sand, silt, clay bedrocks: 1.1, loess regions, fluvial valleys: 1.2, pre-alpine moraines and limestone Alps: 2.5, hills on limestone bedrocks: 3.9 $kmol_c$ $ha^{-1}$ $yr^{-1}$. Calculations of the weathering rates, using an approximation from soil type and texture by Posch et al. [101], give weathering rates of 0.27–2.92 $kmol_c$ $ha^{-1}$ $yr^{-1}$ for the five main weathering rate classes (WRc) for non-calcareous soils in Germany, which agrees quite well with our results. Field weathering rates of base cations in Dutch sandy soils are reported to vary between 0.1 to 0.7 $kmol_c$ $ha^{-1}$ [102–104] cited in van der Salm et al. [105] and 0.16 to 0.58 $kmol_c$ $ha^{-1}$ $yr^{-1}$ [106]. In the Netherlands and Germany, estimated weathering rates for loess ranged from 0.26–1.85 $kmol_c$ $ha^{-1}$ $yr^{-1}$ and 0.350–1.72 $kmol_c$ $ha^{-1}$ $yr^{-1}$, respectively, and in river-clay soils from 0.76–5.3 $kmol_c$ $ha^{-1}$ $yr^{-1}$ [105]. These orders of magnitude are also quite comparable with the data for our model region "loess regions, fluvial valleys". De Vries et al. [7] classified the weathering rates for soil types from unconsolidated rocks in the Netherlands as follows: poor sand: 0.250, moderate poor sand: 0.385, rich sand: 0.520; loess: 0.600; clay: 1.300 $kmol_c$ $ha^{-1}$ $yr^{-1}$.

It should be noted that the rates quoted from the above studies also include Na. They can be related to our values by multiplying by a factor of 0.7 for poor sandy soils and 0.85 for rich soils [107]. However, it is well known that data on weathering can vary widely, and calculations are associated with a high degree of uncertainty [108–112]. Kolka et al. [94] and Wesselink et al. [113] determined uncertainties of 25%. Somewhat

larger uncertainties were found by Jönsson et al. [63] and Dultz [114] with 40% and 75%, respectively. According to Hodson and Langan [115], most methods for determining weathering rates have an accuracy of $\pm 50\%$. Much higher uncertainties of 100% and 250% were reported by Hodson et al. [62,110]. Orders of magnitude above 100% are also given by the works of Klaminder et al. [108] with 98–110% and Futter et al. [109] with 33–300% when comparing different methods for estimating weathering rates. The high variation of weathering rates in the literature can partially be attributed to differences in the methodologies applied, for example different integration levels (soil profile to catchment) or the consideration of gravel content (cf. [108,116,117]). Therefore, when comparing weathering rates determined by different methods, the methodological approach must always be considered.

**Table 2.** Medians of the nutrient balances on the NFI tracts stratified by model regions (Ca and Mg only for carbonate-free soils) for the harvest scenario REAL. DEP: deposition; WEA: weathering; LEA: leaching; HAR: harvest removal; WOH: balance without harvesting; WH balance with harvesting.

| Model Regions | | STOCK | DEP | WEA | LEA | HAR | WOH | WH |
|---|---|---|---|---|---|---|---|---|
| | | | All Tracts | | | | Carbonat-Free Tracts | |
| | | [kmol$_c$ ha$^{-1}$] | [kmol$_c$ ha$^{-1}$ yr$^{-1}$] | | | | [kmol$_c$ ha$^{-1}$ yr$^{-1}$] | |
| Pre-alpine moraines and limestone Alps | Ca | 426.6 | 0.252 | 1.124 | 2.549 | 0.554 | 0.627 | 0.018 |
| | Mg | 134.9 | 0.063 | 1.174 | 1.249 | 0.108 | 0.406 | 0.283 |
| | K | 17.1 | 0.058 | 0.177 | 0.035 | 0.119 | 0.219 | 0.085 |
| Hills on limestone bedrock | Ca | 840.0 | 0.243 | 2.076 | 3.228 | 0.524 | 0.562 | 0.101 |
| | Mg | 75.8 | 0.072 | 1.581 | 0.923 | 0.104 | 0.768 | 0.657 |
| | K | 25.6 | 0.061 | 0.216 | 0.044 | 0.117 | 0.268 | 0.143 |
| Hills on crystalline bedrock | Ca | 46.8 | 0.292 | 0.223 | 0.529 | 0.406 | 0.102 | −0.342 |
| | Mg | 23.9 | 0.087 | 0.295 | 0.471 | 0.091 | −0.021 | −0.128 |
| | K | 11.7 | 0.075 | 0.174 | 0.048 | 0.103 | 0.217 | 0.105 |
| Hills on sand, silt, and clay bedrock | Ca | 172.3 | 0.241 | 0.425 | 0.394 | 0.391 | 0.454 | 0.061 |
| | Mg | 92.7 | 0.076 | 0.402 | 0.322 | 0.088 | 0.215 | 0.122 |
| | K | 18.8 | 0.060 | 0.225 | 0.031 | 0.102 | 0.277 | 0.165 |
| Old moraines, north German lowlands | Ca | 57.2 | 0.201 | 0.136 | 0.099 | 0.240 | 0.298 | 0.045 |
| | Mg | 10.4 | 0.084 | 0.161 | 0.050 | 0.070 | 0.206 | 0.139 |
| | K | 7.2 | 0.071 | 0.050 | 0.026 | 0.065 | 0.110 | 0.043 |
| Young moraines, north German lowlands | Ca | 50.9 | 0.217 | 0.122 | 0.054 | 0.302 | 0.336 | 0.038 |
| | Mg | 6.4 | 0.103 | 0.139 | 0.037 | 0.082 | 0.210 | 0.134 |
| | K | 6.2 | 0.059 | 0.038 | 0.016 | 0.079 | 0.088 | 0.016 |
| Loess region and fluvial valleys | Ca | 275.3 | 0.201 | 0.476 | 0.702 | 0.396 | 0.343 | −0.070 |
| | Mg | 52.9 | 0.073 | 0.522 | 0.340 | 0.077 | 0.232 | 0.147 |
| | K | 17.5 | 0.053 | 0.216 | 0.028 | 0.101 | 0.256 | 0.133 |
| Hills on clay- and silt schist bedrock | Ca | 57.2 | 0.228 | 0.189 | 0.764 | 0.396 | −0.247 | −0.705 |
| | Mg | 32.6 | 0.093 | 0.312 | 0.607 | 0.084 | −0.160 | −0.251 |
| | K | 12.6 | 0.068 | 0.192 | 0.055 | 0.107 | 0.215 | 0.105 |

The median Ca deposition in the stratified model regions ranges between 4 and 6 kg ha$^{-1}$ yr$^{-1}$. Recent results from the Netherlands show similar magnitudes [7]. Except for sites strongly influenced by sea salt deposits and sites with very high precipitation, most sites are characterized by Mg depositions of about 1 kg ha$^{-1}$ yr$^{-1}$. Median K deposition in the model regions varies between around 2 and 3 kg ha$^{-1}$ yr$^{-1}$ (Table 2) and is also quite similar to other studies [7]. The spatial distributions and regional patterns of the deposition input vary largely between the investigated elements (Figure A2). While Mg is strongly influenced by sea salt from the North Sea, this influence is weaker for K. Atmospheric deposits of K are more strongly influenced by local and regional sources [118], which can vary greatly from year to year [46]. In addition to the effect of precipitation and wind speed in the low mountain ranges, the importance of agriculture as a source of K emission is

also evident here. Dämmgen et al. [118] noted that Na and Mg depositions now reached a magnitude that can be described as largely unaffected by anthropogenic factors. In the case of Ca, it is mainly the mountainous areas that show maximum inputs.

Compared to deposition, the leaching rates of Ca and Mg differ more strongly between the model regions (Table 2). In regions with higher fractions of carbonate soils the median leaching losses exceed 50 kg ha$^{-1}$ yr$^{-1}$ (Ca) and 10 kg ha$^{-1}$ yr$^{-1}$ (Mg). For all other upland and hilly areas, the leaching amounts to 8–15 kg ha$^{-1}$ yr$^{-1}$ (Ca) and 4–7 kg ha$^{-1}$ yr$^{-1}$ (Mg). In regions dominated by loess and fluvial valleys, the leaching losses amount to 14 and 4 kg ha$^{-1}$ yr$^{-1}$, respectively. In the poor and moderately poor sandy soils of the regions with "old and young moraine deposits", the leaching losses are much lower and amount to 1.1–2.0 kg ha$^{-1}$ yr$^{-1}$ for Ca and 0.4–0.6 kg ha$^{-1}$ yr$^{-1}$ for Mg. In contrast, K leaching losses are much more homogeneous and differ only slightly between the individual model regions (between 1 and 2 kg ha$^{-1}$ yr$^{-1}$). A similar magnitude for K with a leaching rate of <2 kg ha$^{-1}$ yr$^{-1}$ was also determined in the study of de Vries et al. [7]. Due to the high variability of German soils, a direct comparison of leaching losses with the study of de Vries et al. [7] is difficult, as a larger part of the Dutch forests is located on well-drained sandy soils with sometimes very different nutrient availability. The regions "old moraines" and "loess and fluvial valley" might be the most comparable to the Dutch conditions. The leaching rate for Ca and Mg in the Dutch forests was slightly higher than in our "old moraines" region, at about 4 and 1–4 kg ha$^{-1}$ yr$^{-1}$, respectively, but significantly lower than in the "loess and fluvial valley" region.

The calculated harvest export rates of the sum of Ca, Mg and K ranged from 0.38 and 0.78 kmol$_c$ ha$^{-1}$ yr$^{-1}$ in the different model regions (Table 2). These values are at the upper end of those published in other studies [7,119–123]. It must be emphasized, however, that the current growth rates are higher due to increased N input by deposition [124] and that harvest intensity in many regions of Germany is above the level of many neighbouring countries [25].

*3.2. Nutrient Balances for Different Harvest Intensities*

The base cation content as well as biomass amount differ greatly between the wood, bark, and brushwood biomass compartments. This explains that nutrient export is much more influenced by the harvest intensity than the biomass export.

At the usual harvesting intensity (REAL), about 80% of the total aboveground biomass is removed from the stand, including 6.6% brushwood that accumulates in protective brushwood mats on the skid trails. About 20% of the biomass remains distributed within the stand area as harvest residues. The largest share of harvested biomass (including bark) is stem wood with 55.6%, followed by industrial wood with 13.7%, and the smallest share is fuel wood with 6.9%. The ratio between export with harvest and harvest residues is about 70:30% for the nutrient elements Ca, Mg and K (Figure 4A–C).

The "nutrient preservation potential" of the MIN scenario compared to REAL is mainly related to the assumption that debarking of stem wood and industrial wood is technically feasible during the harvesting process and that the bark remains distributed within the stand area. This assumption is an anticipation of recent developments in harvester technology [125]. In addition, the scenario MIN assumed no accumulation of brushwood on skid trails. When harvesting and timber logging is carried out with forestry machines, brush mats weighing 10–20 kg m$^{-2}$ are required to protect the soil function on the skid trails [126]. Thus, brush accumulation on skid trails could be avoided either by increasing the use of motor-manual harvesting techniques and logging with cable cranes or by consecutive collecting, chipping, and re-distributing brush mats. Both alternatives are considered to be much more costly than conventional fully mechanized harvesting and logging.

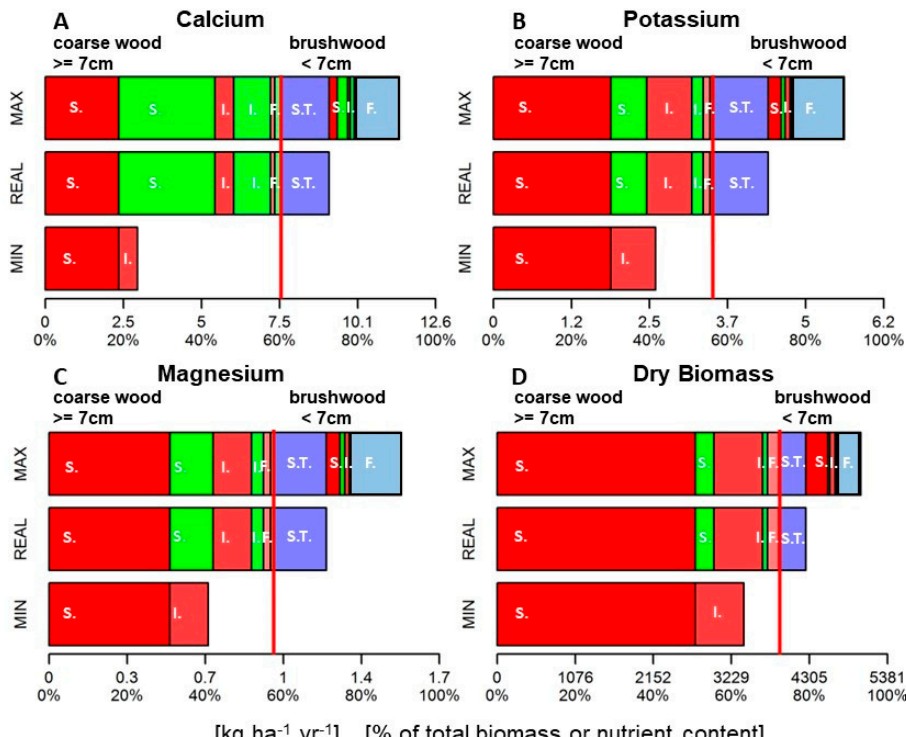

**Figure 4.** Average export of Ca (**A**), K (**B**), Mg (**C**) and biomass (**D**) at NFI tracts with harvested assortments for the three harvesting intensities MIN, REAL and MAX, differentiated into the biomass compartments wood (red), bark (green) and brushwood with bark (blue). Colour intensity decreasing from stem wood over industrial wood to fuel wood. Harvested wood assortments including bark (minimum diameter ≥ 7 cm) marked by a vertical red line. S.: stem wood, I.: industrial wood, F.: fuel wood, S.T.: brush mat on skid trails.

If it would be technically possible to leave all bark and brushwood biomass well distributed in the stand area (scenario MIN), Ca export could be reduced by about 70% compared to the REAL scenario, mainly due to the high Ca content of the bark. Additionally, 43% less Mg and 45% less K are exported in the MIN scenario compared to REAL. Compared to the nutrient loss, the harvest volume of the marketable assortments (stem, industrial and fuel wood) is reduced much less, by about 9% compared to the REAL scenario. Scenario MAX increased the biomass harvest rate by 13% compared to scenario REAL, because wood compartments with diameters < 7 cm (as industrial or fuel wood) are used and commonly occurring harvest losses of 10% are avoided. The average harvest of fuel wood, amounting to only 3.1% in the REAL scenario, could be increased up to 9.7% in the MAX scenario. In addition, stem and industrial wood from recovered harvest losses account for up to 7% of total biomass and could be used in the MAX scenario (Figure 4D). Thus, the fuel wood potential could be extended in this scenario up to 18% of the total aboveground biomass. However, the export for all three nutrients would be about 30% higher in the MAX scenario than in the REAL scenario.

The regional distributions of Ca, Mg, and K balances can reveal hot spots of balance deficits or positive balances and show effects of increasing harvesting intensity (Figure 5). Figure 5 also shows the uncertainty level of the balance calculation at each NFI tract (cf. Section 3.3). Striking regional patterns are found for balance deficits of Ca and Mg, which are most significant in the low mountain ranges with crystalline bedrock and in hill regions with clayey and/or silty shale. The variation caused by harvest intensity is of minor importance for Ca and Mg. The causes are base-poor bedrocks [127] combined with high seepage fluxes [67] and high acid deposition rates [46,128]. Particularly noteworthy here is the hilly 'Sauerland' region, which is characterized by very high conventional harvest intensities [25] and high atmospheric inputs of acidifying components (sulphur

and nitrogen) in the past [45,46]. Akselsson et al. [58] also referred to high historical sulphur deposition and simultaneously high site productivity for their high risk classes. The northern lowlands are dominated by clearly positive Mg balances, and Ca balances also tend to be positive. The pine forests that predominate in this region are generally characterized by low uptake rates of base cations. For example the study of Akselsson et al. [20] indicated that uptake is clearly higher in spruce than in pine. However, the difference is mostly both a species and a site effect [36]. This is compounded by the close proximity of this region to the North Sea and the high deposition rates of base cations by sea spray (see Figure A2).

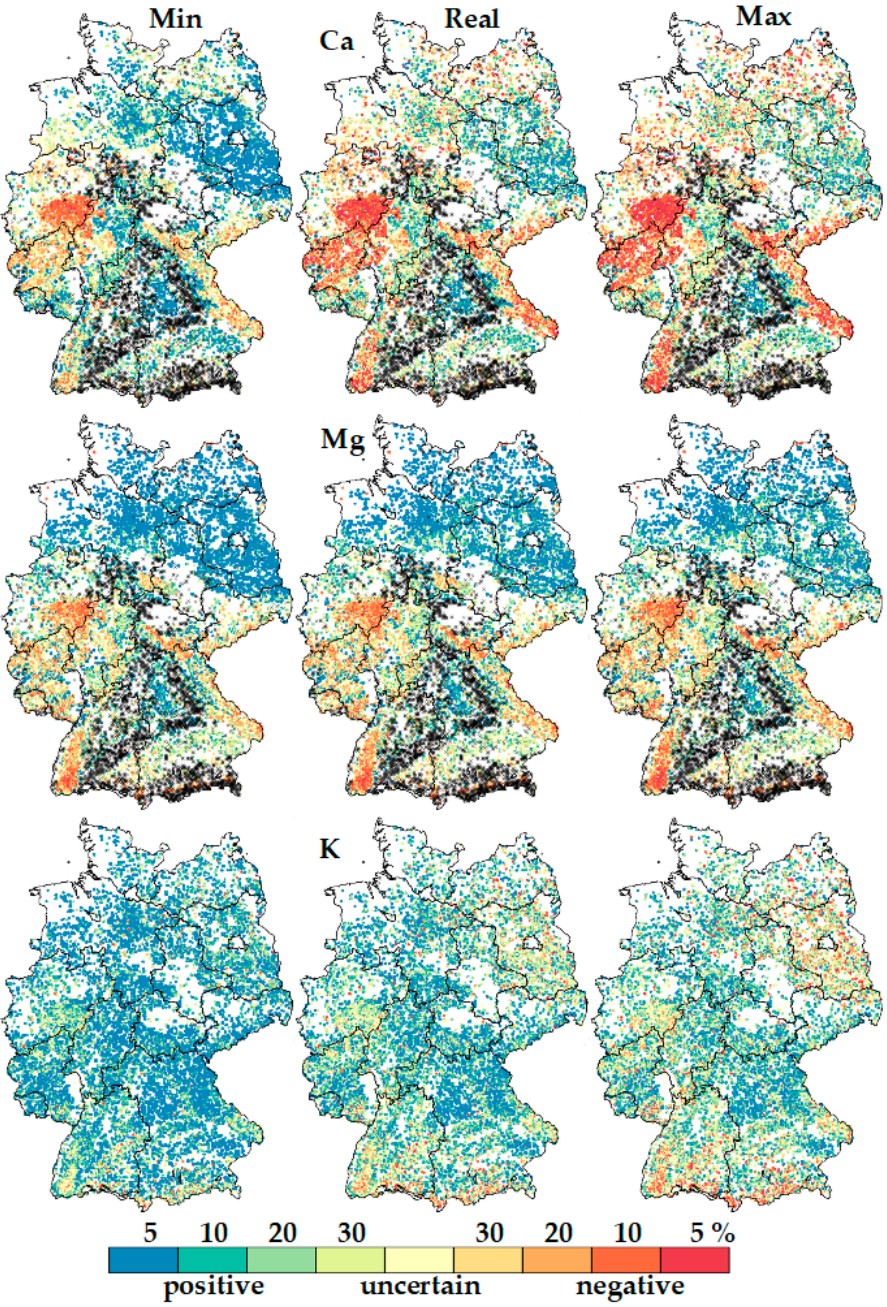

**Figure 5.** Effects of different intensities of biomass use on the nutrient balance of calcium (Ca), magnesium (Mg) and potassium (K) at the NFI tracts (scenario MIN: left column; scenario REAL: centre column; scenario MAX: right column). Balance deficits are shown in red, positive balances in blue. The colour intensity indicates the probability of error.

The K balances show much less pronounced regional differences than Ca and Mg. They are largely positive for the MIN scenario and not significantly different from zero for the REAL scenario. The higher export of biomass under the MAX scenario results in an increase in significant negative K balances for 24.8% of NFI tracts compared to the REAL scenario (18.6%). The K balances tend to be negative in the MAX scenario predominantly in the Alps, the Black Forest, the Swabian Alb, and the sandy sites of the northern lowlands. This is consistent with the results of other studies [7,88,129] and indicates that insufficient K supply may occur on shallow dolomite and limestone soils and on soils with high water permeability (e.g., poor sandy soils).

### 3.3. Uncertainties in the Calculated Balances and Methodological Limitations

When using model results to derive silvicultural management strategies, there is a great risk that policy or forest decision makers may view model results as "absolute" [130]. Therefore, it is important to also assess the uncertainties of modelling and communicate these to users [96,131]. On such a basis, improved decisions can be made, and the limits of model application can be more clearly demonstrated. There are numerous approaches to represent model uncertainties [99,132,133]. In our study, the Monte Carlo method was used because it is very easy to implement and generally applicable to various modelling approaches. The method has been applied to numerous forestry issues and a wide range of topics such as soil acidification [134], critical loads [135], silicate weathering [63], or soil organic carbon stocks [136]. The main disadvantages of the Monte Carlo method [99] are that the results are not available in analytical form and the error can only be determined from a large number of simulations (10,000 in our case) using appropriate statistical settings. Moreover, the multidimensional distributions for correlated variables are often unknown and/or difficult to derive. The latter was considered in the present study by deriving the covariance between the different parts of the nutrient balance on the NFSI plots. However, it should be noted that the derived correlations are affected by large-scale site differences between inventory points. Accordingly, e.g., acidity and regularity of chemical characteristics at the respective soil plots can only be represented to a limited extent. This is impressively illustrated by the relationship between the weathering rate and the element concentrations in the seepage flux/soil solution (see Figure A4). These and the other relationships shown in Figure A4 must be interpreted in such a way that more Ca, Mg, and K is available for leaching at sites with higher soil stocks of base cations and correspondingly higher weathering rates. In addition, the individual members of the nutrient balance were determined independently of each other and, therefore, the simulations could theoretically be carried out without taking the covariance into account. However, the results of the correlation analyses show that the elements of the nutrient balance are not statistically independent random variables and, accordingly, covariance should be considered. Therefore, the models are independent, but the data are not. Accordingly, the presented approach is a compromise that allows a first approximation to the real conditions and uncertainties. Further uncertainties result from the partly incomplete and inhomogeneous data basis for the NFSI points. Particularly problematic is that very important parameters like sulphate ($SO_4^{2+}$), nitrate ($NO_3^-$) and chloride ($Cl^-$) concentrations from 2:1 extracts are not available in some federal states of Germany and where the derived nutrient balances are therefore connected with larger uncertainties (e.g., Brandenburg).

The chosen balancing approach is very vulnerable to large uncertainties in the individual components of the nutrient balance. The simple model of nutrient balancing has so far ignored important interactions between the individual balance components. Such feedback effects particularly influence the already uncertain leaching losses. While the interactions between the cations held on the exchange sites in soils and the concentration of cation in seepage flux are considered in the present approach, the influence of different harvesting intensities on exchanger composition and saturation is not. For example, studies by Zetterberg et al. [137] found reduced exchangeable Ca stocks in the mineral soil and consequently, 40% lower Ca concentrations in seepage water 27–30 years after WTH compared

to conventional harvesting. Paré and Thiffault [39] also discussed the large uncertainties (estimation of element fluxes, lack of feedback effects) in applying the nutrient balance approach as indicator for critical biomass harvesting. Löfgren et al. [138] have criticized the nutrient balance approach for being too uncertain for developing forest ecosystem management strategies and because this approach does not account for all relevant processes. It becomes apparent that the "Simple Mass Balance" approach does not consider key factors of the biogeochemical nutrient cycle and dynamics in forest management, such as changes in humus stocks. Accordingly, nutrient balances are likely to provide a more realistic assessment of the actual situation in short rotation forestry and agricultural land use, as nutrient imports and exports as well as nutrient cycling are easier to assess, with the high proportion of artificial nutrient additions—especially on agricultural land—meaning that interrelated factors internal to the system have a much lower influence on the nutrient balance [39]. The mid- to long-term process dynamics in forest soils would suggest to implement ecosystem process models based on ecosystem models like, e.g., ForSAFE [139], that are however extremely difficult and unsure to parameterize under the influence of fast changing environmental conditions. Therefore, it was decided in this study to confine the balance calculation to the time span of 10 years between two NFI campaigns. Further environmental change can be taken into account by re-calculating the nutrient balances on the basis of data from each new NFI campaign, using the algorithms developed in this study.

*3.4. Effective Options for Nutrient Management*

The balances of the nutrients Ca, Mg and K are influenced by different factors and boundary conditions. Projected changes in soil temperature largely influence the weathering rates of base cations [55]. On the other hand, increasing water limitations may restrict the positive effect of higher temperatures on weathering rates in forest soils as the water content could have a great influence on the weathering rate [62]. Temperature driven increased mineralization rates could lead to higher nitrogen leaching and thus, also to a higher leaching of cations [140]. At present, however, actually the main drivers of base cation leaching are the substantial nitrogen deposition, nitrification processes, and the remobilization of previously retained sulphur in the forest soils [141]. Knust et al. [36] also found for pine and spruce stands in Northeast Germany that Ca and Mg balances became negative due to high leaching rates caused by historically very high sulphur inputs, even with stem-only harvesting. These factors lead to unnaturally high leaching losses of Ca, Mg and K. On the other hand, current forest management strategies can increase or mitigate balance deficits of Ca, Mg and K (Figure 1).

Nutrient export with the harvested biomass can be regulated by forest management, either by adapting the harvest intensity to the vulnerability of the respective forest sites or by technically replacing the exported nutrients by soil protecting liming. Reducing harvesting intensity increases the amount of harvest residuals, which ideally should remain well distributed on the stand area and release nutrients to the soil nutrient pool during mineralization. Through soil protecting liming fine ground dolomite powder or a dolomite-wood ash mixture is applied to the soil surface, and Ca, Mg and K are released within 3–6 years, as the applied materials dissolve slowly [142,143]. Dolomite-wood ash mixture was developed as a new, standardized product for soil protecting liming and has been used in practical forest liming campaigns in the German federal state Baden-Wuerttemberg since 2008 [144]. The wood ash used for this purpose is subjected to strict quality control to ensure that no harmful substances are distributed with it [145]. Thus, both options, adjusting harvest intensity and compensating nutrient deficits through soil protecting liming, allow for ecosystem-compliant replacement of nutrient deficits. In the following, the option of adjusting harvest intensity is derived from the German-wide element balances, incorporating the three elements Ca, Mg, K via Liebig's Law of the Minimum at all approximately 23,000 NFI tracts included in this study (Figure 6).

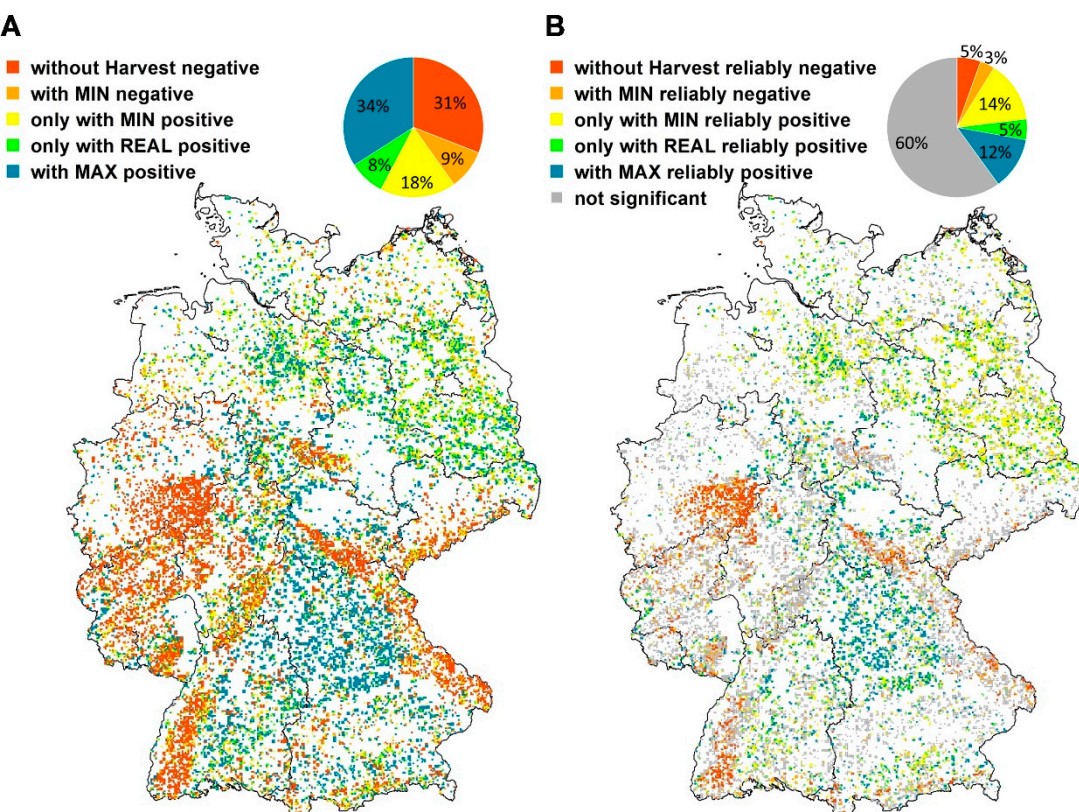

**Figure 6.** Spatial distribution of the base cations (Ca, Mg and K) that can be exported with the harvest scenarios MIN (yellow), REAL (green) and MAX (turquoise) without leading to negative nutrient balance of one or more elements (**A**). On the right map (**B**) the same data analysis is shown, incorporating the uncertainty of the models ($\alpha < 0.1$) to decide about possible harvest intensity. The sites that show negative (or significantly negative) balances without any harvest are marked in red and the sites where scenario MIN shows negative (or significantly negative) balances in orange.

Both maps in Figure 6 represent the same strategy approaches for active nutrient management through regional adjustment of harvest intensity by implementing only those harvesting scenarios that are supported by positive or balanced nutrient balances. If one of the three nutrient balances (Ca, Mg or K) becomes negative, the next more resource-preserving scenario is selected. However, at 31% of the NFI tracts (or at 5% if only the tracts with significantly negative nutrient balances are considered) nutrient balances are negative without any harvest export and therefore, cannot be closed even if harvesting was to completely abandoned. In addition, nutrient balances at 9% of NFI tracts are not large enough to support the lowest harvest intensity (MIN)—and 3% if only significant balances are considered. Satisfactory nutrient management by adjusting harvest intensity to the MIN scenario would be possible at 18% of NFI tracts or at 14% if only significant balances are considered. Conventional use (REAL) can be sustainably realized on 8% (significant on 5%) of the NFI tracts. Including the non-significant balances, the MAX scenario would be possible at 34% of the NFI tracts. If only the significant ones are considered, the MAX scenario is possible at 12% of the sites.

An alternative to reducing the harvest intensity would be to actively manage nutrients by recycling those amounts of nutrients that are required for closing the nutrient balances. The data basis for this approach is the same as for the harvest intensity adjustment. This approach has the advantage of also providing a realistic solution for sites where nutrient balances are negative without any harvesting or have such low positive balances that even the most resource-efficient harvesting scenario (MIN) does not ensure nutrient sustainability.

From our data, we can calculate the area where nutrient management by nutrient recycling is advisable if the present harvest intensity (REAL) is to be maintained. In Table 3, the first two rows show the area with a recycling requirement (based on expected significant nutrient balances only). The last two rows show the annual treatment areas depending on the amount of the yearly balance loss for the deficient nutrients at each tract of the NFI. The repetition time of liming campaigns was derived by cumulating the annual balance deficits until the mean nutrient content of a practical liming campaign with the dosage of 3 tons ha$^{-1}$ dolomite rock powder or 4 tons ha$^{-1}$ dolomite/wood ash mixture was reached, assuming a typical chemical composition of the applied materials. Possible counteracting measures for closing the nutrient balance depend on which nutrient is deficient. Liming with dolomite powder reduces Ca and/or Mg deficiency, whereas at sites with additional K deficiency, a K containing material such as wood ash should be applied in addition to the dolomite. At sites where only K is deficient, a formulation with a high K content should be considered. Summing the annual requirement of liming areas in our model (significantly negative balances for Ca + Mg and Ca + Mg + K), this is an annual requirement of 123,612 ha. This number agrees well with the approximately 100,000 ha yr$^{-1}$ of limed forest area in Germany between 1980 and 2016 [146]. Our 23.6% higher estimate can be explained by the fact that our number refers to the required liming area, but not all forests with a liming demand have been actually limed in the past, and in some federal states no regular liming has been conducted since 1983.

**Table 3.** Areas with balance deficits for element combinations Ca and/or Mg, Ca and/or Mg and K and only K for the total forest area of Germany assuming harvest intensity REAL. Presented are all sites with balance deficits and those where balance deficits are at $\alpha < 0.1$ significant.

| Consideration Level | Unit | Nutrient Element-Combination | | |
|---|---|---|---|---|
| | | CaMg | CaMgK | K |
| Area with balance deficits | ha | 4,274,808 (39.4%) | 1,318,645 (12.2%) | 662,275 (6.1%) |
| Area with balance deficits at $\alpha < 0.1$ | ha | 1,625,254 (15%) | 180,331 (1.7%) | 208,948 (1.9%) |
| Required annual treatment area | ha yr$^{-1}$ | 113,740 (1.05%) | 113,539 (1.05%) | 46,346 (0.42%) |
| Required annual treatment area at $\alpha < 0.1$ | ha yr$^{-1}$ | 84,961 (0.78%) | 38,651 (0.36%) | 41,452 (0.38%) |

Although both discussed management options, adjusting harvest intensity and soil protective liming, can mitigate nutrient loss, it is evident that a combination of both options is required to close nutrient balances at many sites. This particularly concerns those sites that show negative nutrient balances already for the MIN scenario (Figure 5).

Potential consequences of a forest overuse include growth losses [26,27,147] which, among others, would turn short-term gains from fuel wood use into long-term losses. In this context, previous experience with extreme overuse of forests through forest grazing or litter harvesting may be a warning. Furthermore, it should be noted that on sites with low nutrient stocks due to historical overuse, positive balances serve to restore the natural site potential. When using dolomitic lime, complications from nutrient imbalances must also be considered. In this context, special reference should be made to a possible disturbance of the K supply due to calcium-potassium antagonism [148]. Summaries of numerous studies on this topic can be found in Hüttl and Zoettl [149]. Recent evaluations also show that spruce needles have lower P and K levels on limed NFSI sites than on unlimed sites [150]. Corresponding results were also observed on other experimental plots for forest liming [91,151]. Accordingly, the risks of liming to forest ecosystems and potential conflicts with other objectives, like e.g., nature preservation issues should also be included in the decision-making process on nutrient recycling measures [92,152–155].

The evaluations presented here allow a regional estimate of the magnitude of the recirculation requirement and from this also a rough calculation of possible costs. The uncertainty analyses provide the necessary confidence to justify the effort and cost required

for nutrient management measures in practical planning. They also provide a reliable framework for prioritizing measures.

## 4. Conclusions

Sustainable management of forest soils implies that the nutrient export by wood and biomass harvesting, in the long term, does not exceed the nutrient replenishment from weathering and deposition. Based on our findings we recommend distinguishing between sites where negative nutrient balances are mainly due to harvest exports and those that are depleted in nutrients primarily due high atmospheric deposition of sulphur and nitrogen.

Nutrient balances and their uncertainties vary widely depending on geology, soils, climate, deposition history and forest stand characteristics and hence, reliable calculations must be based on local or regional information on those drivers of the nutrient balance. If necessary, the reduction of the harvest intensity can contribute to achieving a sustainable nutrient supply. Alternatively, or additionally, nutrient base cations can be recycled, e.g., by forest liming. However, possible negative side effects must also be considered here. The analysed harvest scenarios show that WTH has a high impact on base cation budgets in forest soils, as WTH removes additional biomass categories (twigs, branches, bark, needles) which have a much higher base cation content than stem wood. This implies a major risk of site degradation, which must be considered in any management. Nevertheless, the option to increase harvest intensity at sites where nutrient balances are significantly positive (12% of NFI sites) allows at least a partial compensation for reductions in harvest intensity at sensitive sites.

The results of this study provide valuable information for practitioners and environmental policy makers to enable spatiotemporal adaptive ecosystem management on the reliable and quality-assured basis of monitoring data. Nutrient balances can be readily adapted to changing environmental conditions by applying the evaluation algorithms developed in this study to data sets from future regular forest monitoring ampaigns.

**Author Contributions:** Conceptualization, B.A. and K.v.W.; methodology, B.A., W.W., C.V., D.Z. and H.P.; validation, B.A., W.W., C.V. and D.Z.; formal analysis, B.A., W.W., C.V. and D.Z.; investigation, B.A., W.W., C.V., H.P., G.K., C.S. and D.Z.; writing—original draft preparation, B.A., K.v.W. and C.V.; writing—review and editing, B.A., W.W., C.V., H.P., G.K., C.S. and D.Z.; visualization, B.A., W.W., D.Z., C.V. and H.P.; supervision, K.v.W., G.K. and C.S.; project administration, K.v.W., G.K. and H.P.; funding acquisition, K.v.W. and G.K. All authors have read and agreed to the published version of the manuscript.

**Funding:** This study was funded by BMEL/FNR (FKZ: 22006512, 22020212 and 22020312). Part of the data were co-financed by the European Commission. Partial funding of data collection and evaluation was provided by the European Union under Council Regulation (EEC) 3528/86 on the Protection of Forests against Atmospheric Pollution, the Regulation (EC) 2152/2003 concerning monitoring of forests and environmental interactions in the community (Forest Focus) and by the project LIFE 07 ENV/D/000218, Further Development and Implementation of an EU-level Forest monitoring Systeme (FutMon).

**Institutional Review Board Statement:** Not applicable.

**Informed Consent Statement:** Not applicable.

**Data Availability Statement:** The datasets related to this article are available from the corresponding author and co-authors on reasonable request. The original datasets for the NFI and NFSI plot and regionalized deposition are available on request from the institutions holding the data. Deposition data by UBA, Dessau, Germany. All NFI and NFSI data by Thünen Institute of Forest Ecosystems, Eberswalde, Germany. Data from German Intensive Forest Monitoring sites (mainly ICP Forest sites) fall under the publication policy of ICP Forests (Annex II of Seidling et al. [156]) and can be accessed from the ICP Forests database (http://icp-forests.net/page/data-requests (accessed on 15 February 2022)). upon request from the Programme Co-ordinating Center (PCC) in Eberswalde, Germany. Deposition calculated by dry deposition factor by Northwest German Forest Research Institute (NW-FVA).

**Acknowledgments:** For the far-reaching provision of data and the close cooperation in all necessary steps of data harmonization, we would like to express our sincere thanks to the representatives of the following state institutions: Research Institute for Forest Ecology and Forestry Rhineland-Palatinate; Hessen-Forest with the Service Center for Forest Management and Nature Conservation (FENA); State Office for Agriculture, Environment and Rural Areas Schleswig-Holstein; Hessian State Office for Nature Conservation, Environment and Geology; State Office for Mining, Energy and Geology Lower Saxony; State Office for Geology and Mining Saxony-Anhalt; Lower Saxony State Forests; Saxony-Anhalt State Forestry Office; State Institute for Ecology, Land Readjustment and Forests North Rhine-Westphalia; State Competence Center Forestry Eberswalde and Forestry Administration of Baden-Wuerttemberg. Maps of the soils and geology of Germany were made available to us by the Federal Institute for Geosciences and Natural Resources (BGR) for use. Data from the monitoring networks of the NFSI and the ICPForest-monitoring (Level II) were provided by the Thünen Institute for Forest Ecosystems in Eberswalde. Germany-wide wet deposition data were provided by the Federal Environment Agency. Further data of great value for the project work were provided by scientists from various research institutes. We would like to thank Hans Pretzsch and Ralf Moshammer (both Technical University of Munich), Joachim Block and Julius Schuck (both Rhineland-Palatinate Research Institute for Forest Ecology and Forestry), and Reinhard Stock (The German Federal Environmental Foundation), who provided extensive data on nutrient concentrations in tree compartments, which were collected in the framework of the project "Decision support Decision Support System for Nutrient Removal in Timber Harvesting" (Ref. 25966-33/0). The constructive comments by the four anonymous reviewers, which helped to improve the manuscript significantly, are gratefully acknowledged.

**Conflicts of Interest:** The authors declare no conflict of interest.

## Appendix A

**Table A1.** Harvesting scenarios used for element balance calculations.

| Scenario | Abbreviation | Description |
|---|---|---|
| Nutrient Saving | MIN | The harvest is limited to saw logs and industrial round wood. The utilization limit varies between top diameters of 12 to 14 cm for softwood and 12 to 17 cm for hardwood species depending on diameter at breast height. Harvest losses remaining on site are assumed to be 10% [33,157]. Brushwood and branch debris are not utilized and remain in the stand interior (not concentrated on the skid trails) through appropriate motorized or mechanized delimbing. In addition, as suggested by Heppelmann et al. [125], debarking of logs takes place inside the stand. This leaves the most nutrient-rich parts (crown material and bark) entirely on site, where they remain available as a source of nutrients. |
| Current harvest intensity | REAL | The main assortments are logs, industrial wood, and fuel wood. All solid volume including bark minus 10% harvest losses is harvested. The utilization limit is set to a top diameter of 7 cm. Harvesting is done by machine and the logging roads are reinforced by half of the arising brushwood volume for soil protection. It is assumed that 80% of the nutrients accumulated in the skid trails with biomass are not available to the system in the medium-term and are thus taken as losses. |
| Highest intensity | MAX | All solid volume is used as log, industrial or fuel wood assuming no technical harvesting losses in the coarse wood fraction. Furthermore, under the same assumptions as in the 'REAL' scenario, half of the produced brushwood is placed on the logging roads. In addition, the other half of the crown material, minus harvesting losses of 20% for conifers and 40% for broadleaved trees, is used for energy production or chemical conversion. This scenario thus largely corresponds to a whole-tree harvesting (WTH). |

## Appendix B

*Estimation of Site Specific Yearly Dry Deposition Factors (DDF)*

To explain the variation in dry deposition, we investigated the relationship between several stand, site and climate parameters and the DDF by generalized additive mixed effect models (GAMM). These models are used for model development taking the "pseudo-replicated" deposition and stand data at every single monitoring site into account. Standard

software to parameterize this type of model is available in the form of the R package "mgvc" [158], which additionally calls for the libraries "MASS" [159] and "nlme" [160]. The model structure is as follows:

$$DDF_{y,p} = b_0 + f_1(x_{1,yp}) + f_2(x_{2,yp}) + \ldots + f_n(x_{n,yp}) + Z_p b_p + \varepsilon \tag{A1}$$

where DDF is the dry deposition factor in year y of plot p, $b_0$ is the intercept term, $f_1$, $f_2$, $\ldots$, $f_n$: are spline smoothing functions, $x_{1,yp}$, $x_{2,yp}$, $\ldots$, $x_{n,yp}$ are $1 \ldots n$ predictor variables of year y of plot p, $Z_p$: is a linear model matrix including dummy variables for coding random effects for plots p with $p = 1, \ldots, 115$. $b_p$ is a vector of random effects and $\varepsilon$ is an independent and identically normally distributed error term.

**Table A2.** Estimated coefficients and statistical characteristics of the dry deposition factor (DDF) model. Est.: estimated parameter value, SE: standard error of the parameter estimates, edf: effective degrees of freedom, $BD_{Na}$: bulk deposition of sodium; WLI: windward/leeward index; WS: wind speed, DN: distance to North Sea; H: stand height; signif. codes: <0.001 \*\*\*, <0.01 \*\*.

|  | Est. | SE | Edf |
|---|---|---|---|
| Parametric coefficients |  |  |  |
| Intercept | 0.23369 \*\*\* | 0.02431 |  |
| Oak | −0.15948 \*\* | 0.05046 |  |
| Spruce | 0.25400 \*\*\* | 0.03381 |  |
| Pine | 0.14753 \*\*\* | 0.04235 |  |
| Approximate significance of smooth terms |  |  |  |
| $\log(BD_{Na})$ |  |  | 2.608 \*\*\* |
| WLI |  |  | 1.000 \*\* |
| WS |  |  | 1.701 \*\*\* |
| DN |  |  | 1.000 \*\*\* |
| H |  |  | 1.000 \*\*\* |
| $R^2_{adj.}$ | 0.648 |  | n = 928 |

To use the model independently from R, the following calculations can be performed:

$$DDF_{y,p} = b_0 + f_1(\ln(BD_{Na})) + f_2(WLI) + f_3(WS) + f_4(DN) + f_5(H) + b_1 \tag{A2}$$

where $b_0 = 0.23369$, and $b_1$ for beech, oak, spruce, and pine is: 0, −0.15948, 0.25400, 0.14753. The different linear and non-linear functions could be calculated as follows:

$$f_1(\ln(BD_{Na})) = 0.0592965032 \cdot (\ln(BD_{Na}))^2 - 0.56333208 \cdot \ln(BD_{Na}) + 0.65236948$$

$$f_2(WLI) = 0.3132540717 \cdot WLI - 0.3559040829$$

$$f_3(WS) = -0.0096723754 \cdot WS^2 + 0.174999468 \cdot WS - 0.5263822623$$

$$f_4(DN) = -0.0015403976 \cdot DN + 0.5616516592$$

$$f_5(H) = 0.0084502178 \cdot H - 0.2362130627$$

where $BD_{Na}$: bulk deposition of sodium [kg ha$^{-1}$ yr$^{-1}$]; WLI: windward/leeward index [–]; WS: annual mean wind speed in 10 m [m s$^{-1}$]; DN: distance to North Sea [km]; H: stand height [m]. In case of negative values, the DDF should be set to zero.

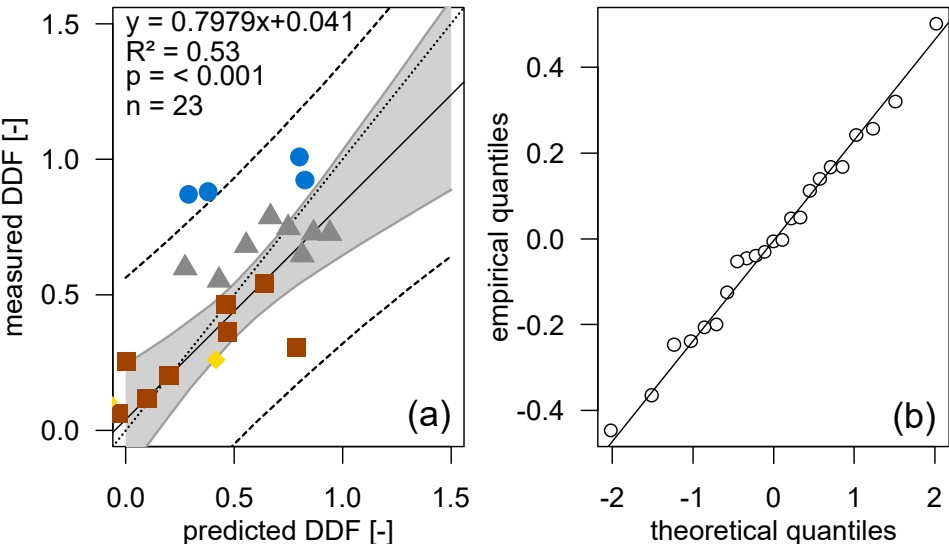

**Figure A1.** Relationship between predicted and measured dry deposition factors (DDF) for forest stands from two different age chronosequence studies in Germany [161,162] used for model evaluation. •: Norway spruce and Douglas fir, ■: European beech, ▲: Scots pine, ◆: Pedunculated/Sessile oak (**a**) Normal probability plot (Q-Q plot) for visual assessment of residuals (**b**).

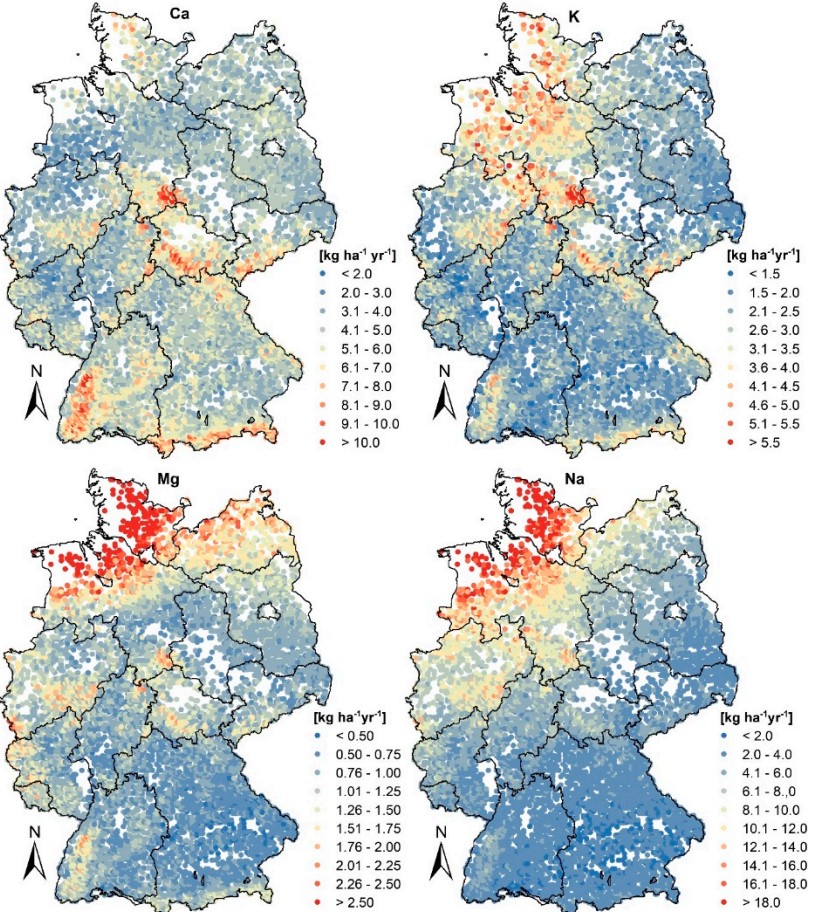

**Figure A2.** Regionalized atmospheric total deposition of the elements calcium (Ca), magnesium (Mg), potassium (K) and sodium (Na) at the tracts of the German National Forest Inventory (NFI).

**Appendix C**

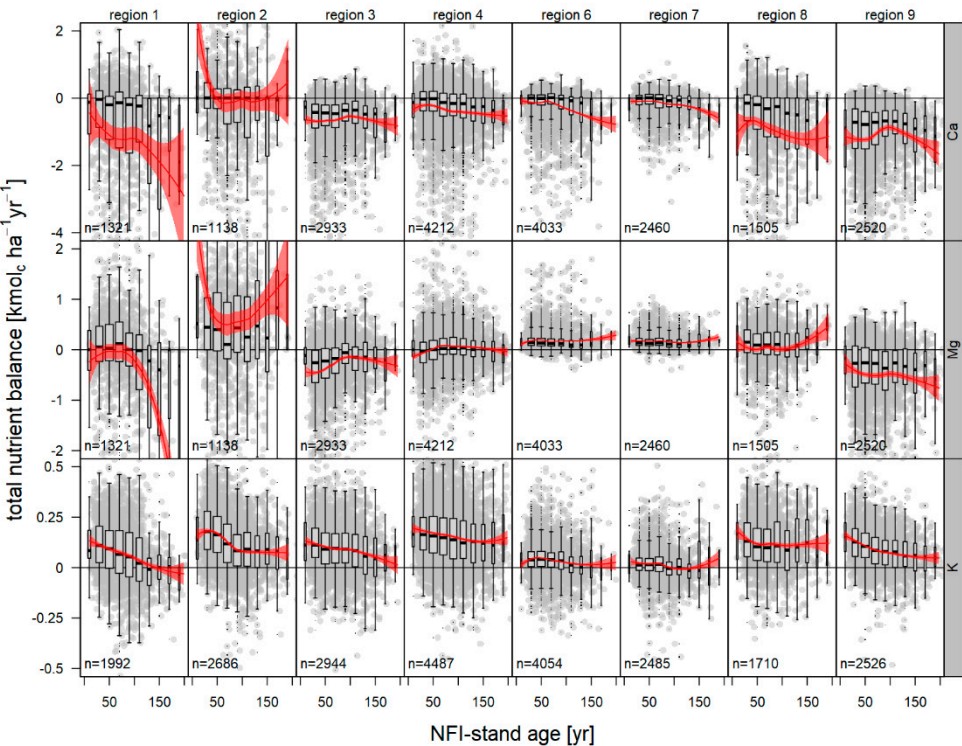

**Figure A3.** Nutrient balance (weathering + deposition—leaching—harvesting removal + soil stock/1000 [scenario REAL]) at NFI tracts over mean stand age at time NFI 2012, stratified by model regions (see Figure 2), for calcium (Ca) and magnesium (Mg) only on carbonate-free soils. Boxplots for age classes of 20 years; the width of the box is proportional to the number of tract corners in the age class; y-axis restricted to 90% of the data, but the data basis for the loess curve (red) is not. 1—pre-alpine moraines and limestone Alps; 2—hills on limestone bedrocks; 3—hills on crystalline bedrocks; 4—hills on sand, silt, clay bedrocks; 6—old moraines, north German lowland; 7—young moraines, north German lowland; 8—loess regions, fluvial valleys; 9—hills on clay- and silt schist bedrocks. Note: the number 5 was deliberately not assigned.

**Appendix D**

*Experimental Design of the Monte Carlo Uncertainty Simulations*

The uncertainty analysis was limited to the carbonate-free sites with respect to the elements Ca and Mg, as the balances for these elements are very uncertain on sites with carbonate soils, but sufficient Ca and Mg supply can be assumed. First, correlation analyses were performed for different variables (see Figure A4 as example from an NFSI plot) used for the derivation of the nutrient balance (deposition, weathering, soil stocks etc.) in the statistical program R using the function rcorr() from the package "Hmisc" [163] to identify possible covariance structures of the parameters in their probability distributions so that they can be taken into account, if necessary. Spearman's correlation coefficients ($r_{Spear}$) were calculated for the correlation analysis because the balance variables, in particular for Ca and Mg were not normally distributed [164]. We only considered significant correlations ($\alpha \leq 0.05$). When non-significant correlations were present, correlation coefficients were set to zero in the further analyses. In addition, mean values and standard deviations for the respective parameters were required. During the Monte Carlo (MC) simulation, random parameter draws were based on the model's expected values and, considering the detected covariance, were realized with the function rmvnorm() from the R package "splus2R" [100]. A compilation of these values (the mean values are site-specific and therefore cannot be presented in the Table A3) is provided in Table A3 for the NFSI grid points. Table A4

contains the parameters to characterize the uncertainties of the regionalization models (see Section 2.3) for the NFI tracts.

**Table A3.** Input data and uncertainty ranges of the balance variables as the basis for the Monte Carlo simulations of the soil balances components on the NFSI points. DEP: deposition; WEA: weathering rate; CSE: concentration in seepage water; SEF: seepage flux; STO: soil stock of plant available nutrients; mean µ: site mean; min: minimum value; SD: standard deviation; RMSE: root mean square error; R: data reference.

| Parameter | Unit | Mean µ | Min | SD[%] | RMSE | R |
|---|---|---|---|---|---|---|
| $DEP_{Ca}$ | kg ha$^{-1}$ yr$^{-1}$ | modelled results | 0.001 | - | 2.68 | [165] |
| $DEP_{Mg}$ | kg ha$^{-1}$ yr$^{-1}$ | modelled results | 0.001 | - | 0.43 | [165] |
| $DEP_{K}$ | kg ha$^{-1}$ yr$^{-1}$ | modelled results | 0.001 | - | 1.54 | [165] |
| $WEA_{Ca}$ | kg ha$^{-1}$ yr$^{-1}$ | modelled with site specific data | 0.001 | 23.3 | - | [166] |
| $WEA_{Mg}$ | kg ha$^{-1}$ yr$^{-1}$ | modelled with site specific data | 0.001 | 23.3 | - | [166] |
| $WEA_{K}$ | kg ha$^{-1}$ yr$^{-1}$ | modelled with site specific data | 0.001 | 23.3 | - | [166] |
| $CSE_{Ca}$ | mmol$_c$ l$^{-1}$ | modelled with site specific data | 0.001 | 53.9 | - | [73] |
| $CSE_{Mg}$ | mmol$_c$ l$^{-1}$ | modelled with site specific data | 0.001 | 57.5 | - | [73] |
| $CSE_{K}$ | mmol$_c$ l$^{-1}$ | modelled with site specific data | 0.001 | 97.1 | - | [73] |
| SEF | mm | modelled with site specific data | 0.001 | 8.3 | - | [65] |
| $STO_{Ca}$ | 0.001 kg ha$^{-1}$ | measured | 0.001 | 50.0 | - | [167] |
| $STO_{Mg}$ | 0.001 kg ha$^{-1}$ | measured | 0.001 | 50.0 | - | [167] |
| $STO_{K}$ | 0.001 kg ha$^{-1}$ | measured | 0.001 | 50.0 | - | [167] |

**Table A4.** Coefficient of variation (based on RMSE of the validation) of the regionalized balance elements for the individual model regions and the global model as a basis for the Monte Carlo simulations of the soil balances on the NFI tracts. Model Region (see Figure 2A, main text): 1: pre-alpine moraines and limestone Alps; 2: hills on limestone bedrocks; 3: hills on crystalline bedrocks; 4: hills on sand, silt, clay bedrocks; 6: old moraines, north German lowland; 7: young moraines, north German lowland; 8: loess regions, fluvial valleys; 9: hills on clay and silt schist bedrocks. Note: the number 5 was deliberately not assigned. X: Global model, *: Global model used (cf. Section 2.3).

| Model Region | Leaching | | | Weathering Rate | | | Soil Stocks | | |
|---|---|---|---|---|---|---|---|---|---|
| cf. Figure 2A | Ca | Mg | K | Ca | Mg | K | Ca | Mg | K |
| 1 | 87.9 * | 125.7 * | 80.5 * | 321.8 * | 89.2 * | 55.8 * | 72.7 * | 80.1 * | 51.9 * |
| 2 | 57.3 | 106.9 | 76.8 | 109.8 | 153.8 | 50.8 | 57.7 | 127.3 | 48.6 |
| 3 | 77.5 | 83.3 | 79.1 | 90.7 | 87.2 | 55.8 * | 139.9 | 126.8 | 55.1 |
| 4 | 103.8 | 105.8 | 69.7 | 321.8 * | 132.8 | 54.5 | 110 | 107.1 | 60.2 |
| 6 | 80.8 | 95.8 | 107.6 | 107.5 | 73.4 | 78.6 | 117.9 | 127.6 | 56.6 |
| 7 | 102.9 | 96.8 | 105.1 | 321.8 * | 75.9 | 65 | 71.6 | 103.2 | 51.7 |
| 8 | 80.6 | 76.1 | 81.3 | 47.8 | 47.9 | 51.1 | 89.1 | 86.5 | 50.1 |
| 9 | 119.2 | 125.7 * | 95 | 321.8 * | 261 | 65 | 112.2 | 65 | 136.8 |
| X | 87.9 | 125.7 | 80.5 | 321.8 | 89.2 | 55.8 | 72.7 | 80.1 | 51.9 |

Again, the relative root mean square error (RMSE) was used as an indicator for the (unobserved) standard deviation, as the models were fitted with log-transformed data. If only information on the maximum error range (e.g., ±25% for seepage flux) was available,

the percent standard deviation σ was determined as follows (for the normal distribution, 99.7% of the values are in the ±3-σ range):

$$\sigma = (\text{error})/3 \qquad [\%] \qquad\qquad (A3)$$

Figure A4 shows an example of the distribution of randomly selected values for the NFSI plot with the ID 30016 and the balance variables deposition, weathering, nutrient concentration in seepage water, seepage flux, and soil stock of Ca.

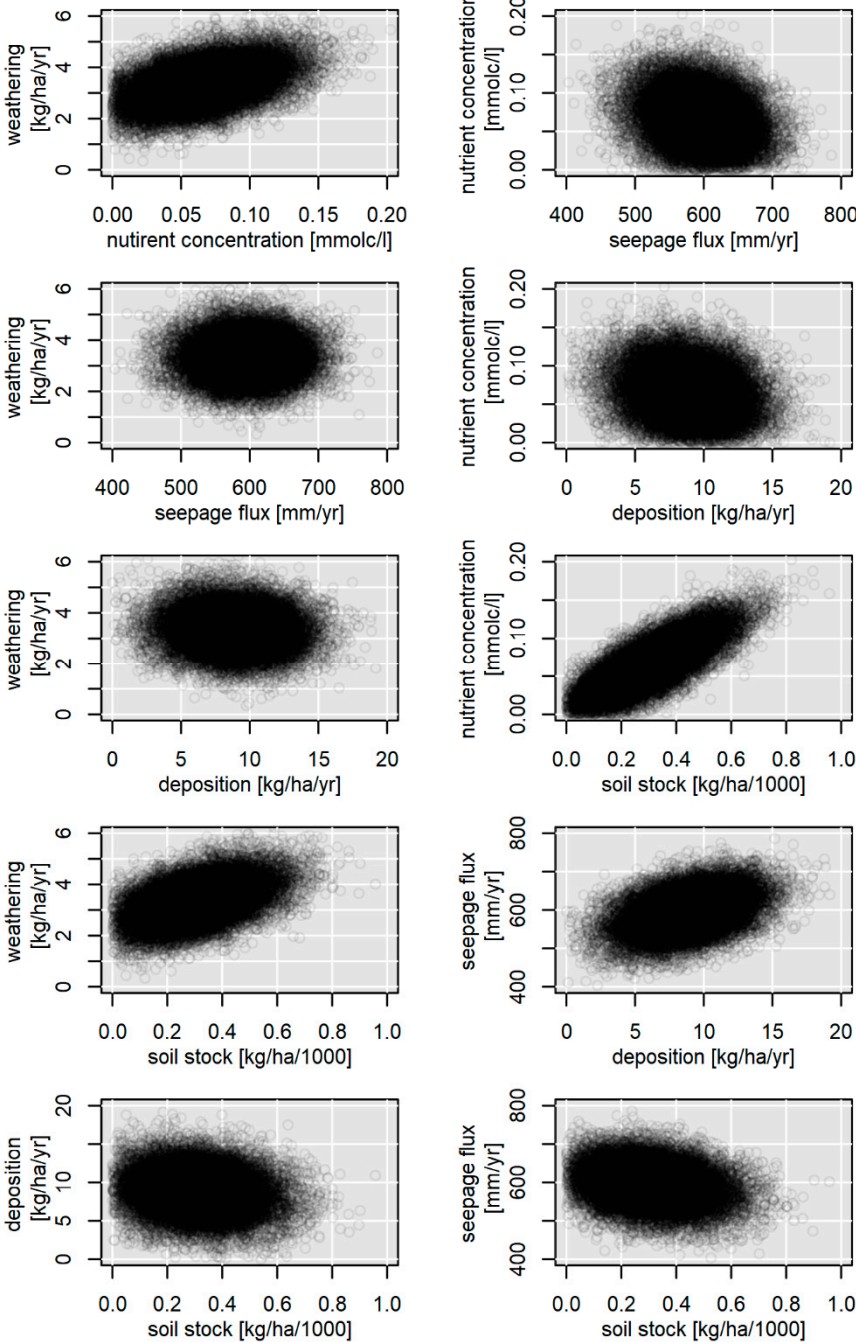

**Figure A4.** Example for randomly varied input data of the calcium (Ca) balance considering the covariance in the Monte Carlo procedure for the NFSI plot with the ID 30016.

As the function rmvnorm() does not offer the possibility to represent truncated parameter spans, but negative balance variables are implausible, 50,000 simulations were first

performed for each NFI tract. From this data set, all simulations were used for which the balance variables reached values $\geq 0.001$. Finally, a random sample ($n = 10{,}000$, [168,169]) was generated from this subset using the sample() function without replacement. Such a sample size allows the necessary reproducibility of the obtained results [63].

Except for deposition and harvesting removal, the calculated balance elements at the NFI sites exhibit—besides a methodological (model-related) error—an additional regionalization error. For deposition and harvest, the estimated model error at the NFI tracts was used. The example of weathering rates (WEA) is used to illustrate how the calculation of the i-th value within the Monte Carlo simulations $\left(\text{WEA}_i^{\text{MC}}\right)$ is based on the regionalized, site-specific value of $\text{WEA}_{\text{STA}}$, the regionalization error $\left(\text{WEA}_{i,\text{REG}}^{\text{MC}} - \text{WEA}_{\text{STA}}\right)$, and a methodological error $\left(\text{WEA}_{i,\text{MET}}^{\text{MC}} - \text{WEA}_{\text{STA}}\right)$. The terms $\text{WEA}_{i,\text{REG}}^{\text{MC}}$ and $\text{WEA}_{i,\text{MET}}^{\text{MC}}$ are the simulation terms drawn during the Monte Carlo simulation from the corresponding distribution of the regionalization and from the modelling derived weathering rates. Thus, the distribution of simulated weathering rates is generated as follows:

$$\text{WEA}_i^{\text{MC}} = \text{WEA}_{\text{STA}} + \left(\text{WEA}_{i,\text{REG}}^{\text{MC}} - \text{WEA}_{\text{STA}}\right) + \left(\text{WEA}_{i,\text{MET}}^{\text{MC}} - \text{WEA}_{\text{STA}}\right) \qquad \text{(A4)}$$

The error in harvest removals was accounted for in a slightly different manner. The calculation of element removals is based on forest growth simulations, biomass functions and compartment specific nutrient levels and, therefore, very time consuming and not feasible to implement for 10,000 Monte Carlo iterations. Instead, for each NFI tract, the corresponding harvest removal was calculated, considering the error propagation of the prediction errors. Only the errors of the biomass functions and the models for nutrient levels in biomass were taken into account. Other elements, such as the error of the model WEHAM in predicting future forest development and of BDAT (program to calculate e.g., diameters, volume, assortments and double bark thickness for different tree species based on tree characteristics and sorting information [77]), as well as uncertainties in the input data for predicting nutrient levels were ignored. The time required for repeated evaluation of these models (updating of diameter at breast height, tree height and growth, possibly changing the dropping out collective and variety composition, re-evaluation of proximity element functions with varying predictors) is not proportionate to the expected gain in accuracy. In addition, the error rate in the estimation of biomass and approximate elements is much lower than the uncertainties of the other nutrient balance elements.

Many of the required error specifications could be derived directly from model validation of the various sub models. Assuming that the errors in the statistical models are normally distributed, the RMSE (root mean square error) was simplified to be equal to the standard deviation. For models with heteroscedastic errors (biomass models and log models of regionalization), the coefficient of variation based on RMSE, i.e., relative RMSE calculated as RMSE/E(Y) (E: expected value), was used.

For some parameters, data from the literature were used for a rough estimation. For soil nutrient stocks a coefficient of variation (CV) of 50% was assumed (cf. Table A3).

After the simulations, the significance level for the occurrence of negative or positive element balances was determined from the probability densities of the 10,000 Monte Carlo realizations. The terms significant and weakly significant are defined with error probabilities of $\alpha \leq 0.05$ and $0.05 < \alpha \leq 0.1$, respectively. Figure A5 shows this procedure as an example for three situations. In example B1 (red distribution curve), the distribution of the 10,000 simulations is mainly negative, but indeed encompasses also zero. More importantly, zero ranges between the 90% and 95% percentile (highlighted in red). Accordingly, the balance for this example is only weakly significantly negative ($0.05 < \alpha \leq 0.10$). Example B2 in Figure A5 shows slightly more positive than negative realizations, but this is not statistically significant: (the area between the 30% and 70% percentiles (highlighted in colour) includes zero). Example B3 in Figure A5 shows a situation of significantly positive ($0.05 < \alpha \leq 0.1$) balance simulations. Here, the value zero is excluded with over 95%

confidence (the 5% quantile of the distribution is highlighted in blue). Results (Figure 5, main text) were plotted using the R package "maptools" [170].

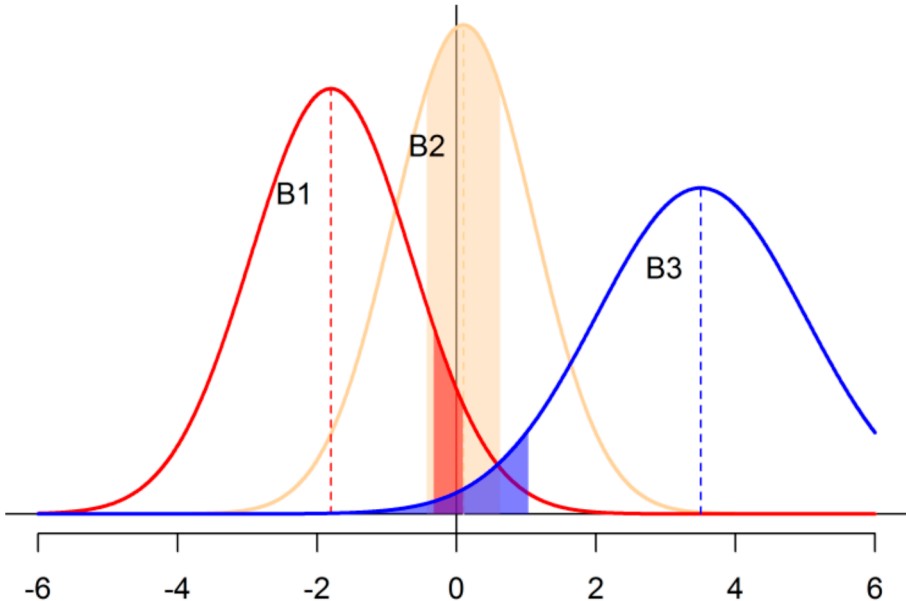

**Figure A5.** Examples (B1 to B3, see text) for deriving significance levels of negative or positive soil nutrient balances.

## Appendix E

*Validation of the Regionalized Nutrient Balances*

The regression models developed at the test dataset of NFSI were applied with the predictors at the validation data set and compared with the measured NFSI data at each plot of the validation data individually. The assessment error of Ca and Mg balances were high (RMSE = 62.5 kg ha$^{-1}$ yr$^{-1}$ and RMSE = 23.2 kg ha$^{-1}$ yr$^{-1}$, respectively), whereas it was low for K (RMSE = 4.3 kg ha$^{-1}$ yr$^{-1}$). The relation between the balance values calculated from measured NFSI data and regionalized data at NFI sites follows closely the 1:1 line as it is presented in Figure A6 exemplarily for Ca and K.

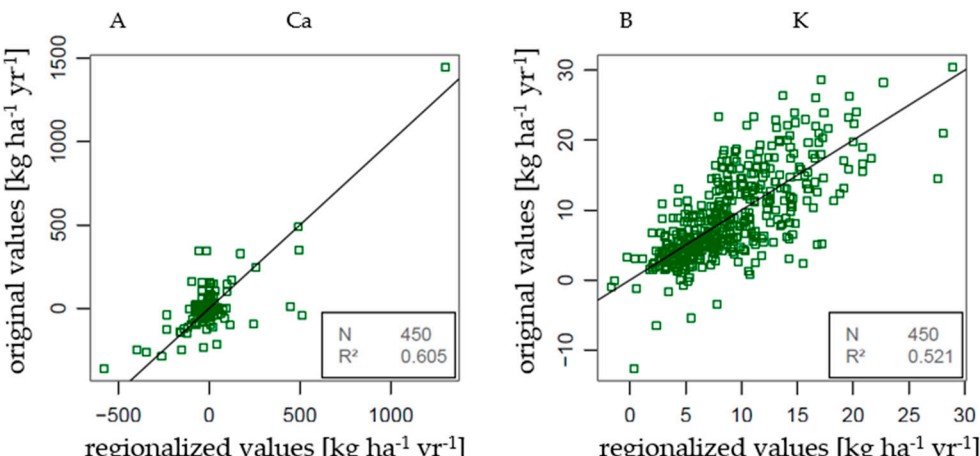

**Figure A6.** Scatter diagram of regionalized calcium (Ca, (**A**)) and potassium (K, (**B**)) balances versus balances calculated from measured NFSI data (*y* axis). Black line = 1:1 reference.

The spatial distribution of Mg balances calculated from measured NFSI data and regionalized Mg balances at NFI tracts is presented by the Figure A7. The comparison of the spatial distribution of Mg balances between those calculated with measured NFSI data

and the regionalized balances in NFI sites demonstrates impressively that the regionalized data provide 26 times higher information density in space and much more consistent regional patterns of balance levels. It is also visible that regions with lacking NFSI data for balance calculation are supplemented by the regionalization process in a sensible way.

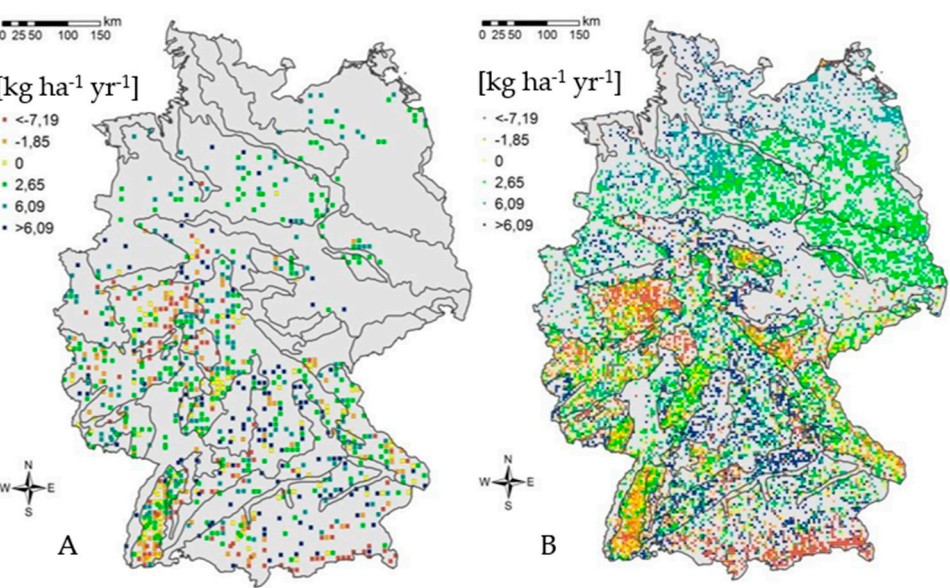

**Figure A7.** Spatial distribution of Mg balances calculated with measured NFSI data (**A**) and regionalized Mg balances at NFI sites (**B**). Black lines are boundaries of model regions (see Figure 2).

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
