# Peer review of "Merits and Limitations of Element Balances as a Forest Planning Tool for Harvest Intensities and Sustainable Nutrient Management—A Case Study from Germany"

_soilsystems, doi:10.3390/soilsystems6020041_

Round 1

Reviewer 1 Report

This manuscript describes a model to determine nutrient balances in Germany due to traditional or whole tree harvesting.  This manuscript is exceptionally long and wordy.  One option is to focus on the models used to develop the nutrient balances for one manuscript and use these models in a second manuscript to describe the effects of harvest intensities.  Another option is to severely cut down the descriptions of the methods used in this manuscript.  At 42 pages, this manuscript reads more like a book than an article.  Consider using more figures instead of listing numbers in the text.  I think that this is a very useful manuscript for determining the effects of harvest intensities.  I think it should be reconsidered after major revisions.

Lines 42-44: This sentence is confusing as written.

Line 155: “as database” needs to be fixed.

Line 197: I think you mean section 2.1.

Lines 304-306: Consider rewriting this sentence.

Lines 415-416: Do you mean that liming was not considered in these calculations?  I feel like the section on liming could be removed or shortened.

Lines 430-450: This uncertainty analysis discussion could be shortened to what was done for your analysis.

Line 452: “known”

Lines 489-498: Consider making these values into a graph. Or, it looks like weathering rates are in Table 3 so they can be removed from the text.

Figure 4. This figure needs more clarification.  Why are there multiple bars of the same color?

Figure A1 is missing.

Author Response

Reviewer 1 – Reply

Reply: Dear Reviewer, thank you for your comments which are very helpful for revising and improving our manuscript. Since the task to determining “the effects of harvest intensities” that you yourself quoted as “very useful” is a rather complex item and needs competence in very diverse skills (soil chemistry, soil physics, soil mineralogy, water flux- and element flux modelling, deposition modelling, geo statistics, error analysis, yield science etc.) that is not common to the average reader in full extent, we decided to choose option 2 you suggested. Therefore we shortened the introduction and methods chapter but intended to preserve a comprehensive overview over the different methods used. We are convinced that splitting the method development and the model application would deteriorate the traceability of the scientific procedure and also the option for the readers to judge the reliability of the model results. We shortened our manuscript in two ways. Firstly we have shortened many sections or deleted them altogether and secondly we have moved some passages that we found to be too differentiated in the main text but necessary for in-depth understanding of interested readers, from the methods chapter to the appendix. Additionally, we have also significantly shortened the chapters on results and discussion, so that hopefully the work now reads more like an article and not “like a book” as well, as you rightly pointed out. These measures made it possible to reduce the main text in a volume of 3 pages. We hope that this way the results will be better presented.

Specifically, we made the following changes:

  • We deleted the subtitles in the introduction
  • The introduction was shortened
  • sections 2.1 and 2.2 have been combined into one, which was additionally shortened
  • The original section “Fundamentals of nutrient balancing” was shortened
  • We have moved Table 1 with the exact and detailed definitions of the harvesting scenarios to the Appendix A.
  • Some cuts in the former sections 2.2.3, 2.2.4, 2.2.5, 2.2.6, 2.3, 2.4
  • We shortened in the range from line 430-450
  • We shortened the section “Treatment of special sites” especially the subsection of liming
  • We have significantly shortened chapters 3.1 to 3.4 for a better presentation of the results.
  • In many places of the text we have improved the English.

Minor comments

Lines 42-44: This sentence is confusing as written.

Reply: We agree and have reworded the sentence.

Line 155: “as database” needs to be fixed.

Reply: We agree and have changed it to: “as spatial data basis”

Line 197: I think you mean section 2.1.

Reply: We agree. Based on another reviewer's comment, we have rewritten this sentence.

Lines 304-306: Consider rewriting this sentence.

Reply: We have rewritten this sentence.

Lines 415-416: Do you mean that liming was not considered in these calculations?  I feel like the section on liming could be removed or shortened.

Reply: Exactly: that is what we mean. We follow your suggestion and have shortened the section on liming

Lines 430-450: This uncertainty analysis discussion could be shortened to what was done for your analysis.

Reply. We followed your suggestion and shortened the range from line 430-450.

Line 452: “known”

Reply: Thank you. Known is right! We changed it

Lines 489-498: Consider making these values into a graph. Or, it looks like weathering rates are in Table 3 so they can be removed from the text.

Reply: you are absolutely right that the weathering rates are given in Table 3. But these rates are element specific. To allow the reader an easier comparison with the following literature data in the discussion we have summed up the element specific values. Therefore, we have listed the values in the text. Since the basis of the calculations is already listed in Table 3, we do not consider that an additional table is necessary. Accordingly, we would like to leave these values here in the text to facilitate the discussion for the reader.

Figure 4. This figure needs more clarification.  Why are there multiple bars of the same color?

Reply: We agree. We inserted additional litters in the Figure (S. = Stem wood; I. = Industry wood; F. = Fuel wood; S.T. = brush mat on skid trails.) for more clarification.

Figure A1 is missing.

Reply: Figure A1 was still present in the submitted Word document. There must have been a problem with printing the word file to the pdf document. Note Fig. A1 is now B1.

Reviewer 2 Report

Very intersting synthetic manuscript, with very topical ideas. It is very complexe and no all aspects are in my conpetence range. Introducing it has be emphasized also, that reviewer is not native speaker and my statement, that manuscript (language) is well understandeable, can be nor so relevant from the point of view of native English speaker.

Several remarks and comments:

The research is oriente dat basic cations, there is clear, that nitrogen dynamics is different. The problems connected by phosphorus areworth to analyze, but I respect menaning of authors, in the future, the P problems has to be studied too.

Fig 1: Harvesting – text correction

Page 297: resulting data „was“ or „were“?

The results are useful for general forest management at a national scale. It i salso a flexible decision supporting tool in changing conditions.

Author Response

Review 2 – Author Response

Reply: Dear reviewer, many thanks for taking your time to read our manuscript and for you helpful input.

R2: The research is oriented at basic cations, there is clear, that nitrogen dynamics is different. The problems connected by phosphorus are worth to analyze, but I respect meaning of authors, in the future, the P problems has to be studied too.

Reply: P is really a very big problem for future investigations and greater efforts need to be made to balance it better for forest ecosystems.

Fig 1: Harvesting – text correction

Reply: Thank you! We corrected the text in Figure 1!

Page 297: resulting data „was“ or „were“?

Reply: we changed it to were.

The results are useful for general forest management at a national scale. It is also a flexible decision supporting tool in changing conditions.

Reply. Thank you!

Reviewer 3 Report

The manuscript entitled "Merits and limitations of element balances as a forest planning tool for harvest intensities and sustainable nutrient management - a case study from Germany" deals with the important problem of element pool depletion in soils due to timber harvesting.  The authors undertook the difficult task of assessing the calcium potassium and magnesium balance in German forest soils. This is a pioneering approach to the problem at this spatial scale. Modelling methods were used as the only currently possible way to answer the research task. It can be seen that the authors apply various modelling methods with great freedom and that they understand the purpose of the methods used. They avoided a common mistake in which methods become more important than the purpose of work. They are also aware of the limitations of the applied methods both due to the limitations of the mathematical apparatus itself as well as due to the variability of weathering processes, leaching and intensity of harvesting. The paper is well written and, despite a very detailed discussion of the literature, clearly leads the reader behind the main idea. Conclusions of the paper are balanced and important both from a scientific point of view as well as being a valuable guide for forest practitioners.

I believe that the presented article deserves to be published in the journal Soil System.

Author Response

Review 3 - Reply

Reply: Dear reviewer, thank you for your time to read our manuscript and your positive feedback.

Reviewer 4 Report

In the revised article, submitted to Soil Systems MDPI by Bernd Ahrends and Co-authors have presented results of interesting analysis on soil parameters based on large scale inventory conducted in Germany.  Generally, I find the manuscript valuable as the study concerns 1) soil chemistry and forest management, 2) large data set at landscape scale. In my opinion, this article may be interesting for soil scientists, ecologist and may give the background to extend ecological studies. The manuscript is generally complete and well prepared, but some corrections need to be done before publication. Truly say I am not used to correct the manuscript where results and discussion is joint, but in some aspect the manuscript gains when they (results) are described and explained. The manuscript should be shortened, in introduction and the subtitles 1.1 etc may be omitted. Moreover, figures lack letters which makes navigation in manuscript difficult. Some values from the text should be checked and corrected. Additionally, there are statements which may be omitted (line 646) and I suggest changing “comparable” to similar, equal etc. Minor (mainly editorial) comments are also presented in attached PDF file.

Author Response

Review 4 - Reply

Reply: Dear reviewer, thank you for your time, suggestions and for numerous minor comment. We have shortened the manuscript at numerous places in two different way. Firstly we have shortened many sections or deleted them altogether and secondly we have moved some passages that we found to be too differentiated in the main text but necessary for in-depth understanding of interested readers, from the methods chapter to the appendix. You have noticed that it is unusual to you that we merged results and discussion. We have done this deliberately to make it easier for readers to follow the study. We are convinced that it would be favorable to discuss the results directly following their presentation. In this way, the context becomes clearer with respect to the above-average complexity of the item of our study. The alternative of placing all discussion parts in a separate section would require many additional explanatory sentences and thus would further stretching the text, which we wanted to avoid. We agree that: “in some aspect the manuscript gains when they (results) are described and explained”. For this reason, we have also significantly shortened the chapters on results and discussion. These measures made it possible to reduce the main text in a volume of 3 pages. We hope that this way the results will be better presented.

Specifically, we made the following changes:

  • In the introduction we omitted the subtitles.
  • The introduction was shortened
  • sections 2.1 and 2.2 have been combined into one, which was shortened
  • The original section “Fundamentals of nutrient balancing” was shortened
  • We have moved Table 1 to the Appendix.
  • Some cuts in the former sections 2.2.3, 2.2.4, 2.2.5, 2.2.6, 2.3, 2.4
  • We shortened the description of the uncertainty analysis (Lines 430-450)
  • We shortened the section “Treatment of special sites”.
  • We have significantly shortened chapters 3.1 to 3.4
  • We have revised the conclusions
  • We have integrated letters (A,B,C and D) in the figures
  • We have checked and corrected some values
  • We followed your suggestion and omitted the problematic statements (e.g. line 646)
  • We changes the wording “comparable” to similar.
  • We followed all your suggestions in the succeeding minor comments

Minor (mainly editorial) comments

Line 138, 139 and 143: I suggest using full elements name at the beginning of the sentence.

Reply: we changed it.

Line 152: Please improve text resolution on fig 1

Reply: we improved the text resolution on Figure 1

Line 172: Please provide letters A, B here and on the fig 2

Reply: We provided the reference with additional letters in the Figure 2

Line 174: Which exactly?

Reply: We have updated the reference to Figure 2B

Line 178: This abbr. was explained previously. Please remove.

Reply: We removed it

Line 183: Please provide scale bar and North direction on figure and letters A and B

Reply: we inserted a scale bar and a North arrow in the map and the letters A and B

Line 196: This part may be omitted.

Reply: We have deleted the half sentence!

Line 234: Please add to reference list

Reply: we added it to the reference list

Line: 272: ..in....Please add name before the number

Reply: We added the name “Weis et al.” before the number

Line: 293: This abbr. was previously presented. Please remove.

Reply: we removed the abbreviation

Line: 298: This is the aim of the study and should be presented at the end of the introduction.

Reply: We absolutely agree. We deleted the sense in this chapter. In the instruction this aim has been already mentioned as the first main research question.

Line 362: Sometimes authors use Fig. not Figure. Please correct.

Fig 2 a or 2B

Reply: We corrected all Fig. to Figure. Additionally we added the letter A to the reference Figure 2.

Line 484: Please provide letters A, B, C and use them on the text.

Reply: We have inserted the letters A, B and C into the figure.

Line 490: Fig or Figure?

Reply: We corrected all Fig. to Figure. Thank you!

Line 537: is between..... and ...... Please check and correct.

Reply: We agree and corrected this.

Line 538: Please change to similar or equal as many things are comparable but this not carry any information.

Reply: we changed to “similar”.

Line 541: Please see comment from line 537

Reply. We agree and corrected this.

Line 542: See comment in line 538

Reply: We changes to “similar”.

Line 542 and 532: Please avoid such kind of statements. I suggest reporting data and citing Appendix A2 in brackets.

Reply: we have deleted the corresponding sentence with the statements and we cited “Appendix A2” in brackets in the following sentence.

Line 559: Do you mean "difference"?

Reply: Yes, that was exactly what we mean. We changed to difference

Line: 562(2 times): Please, check previous comment in line 537

Reply: We agree and corrected this.

Line 563: There is a lack of the verb.

Reply: We agree and have completed the sentence.

Line 567: Please, check previous comment in line 537

Line 568: similar?

Reply: We agree and changed to “similar.

Line 578: Please remove Tab. 3 from brackets

Reply: We removed Tab. 2 from the brackets and changed Tab. To Table.

Line 579: ...regions (Tab. 3).

Reply: We changed, see also the comment above.

Line 597: Please add letters A, B, C, D

Reply: at this place we have inserted the letters A-C. Thank you

Line 612: Please provide letters A, B, C, D.

Reply: we have inserted the letters. A, B, C and D in Figure 4

Line 638: Please, remove this dot [.]

Reply: We have removed this dot

Line 646: Please, provide data and avoid describing what is presented on figures

Reply: We agree and have deleted the sentence completely.

Line 669: All results and data are comparable. Please provide: similar, lower, higher.

Reply: We agree and changed to “similar”.

Line 783-785 Please check previous comment. Provide data/summary/comment without describing what is on the figure.

Reply: We agree and we have rearranged the sentence accordingly

Line 805: Please provide A and B

Reply. We inserted the letters A and B in the Figure 6

Line 820: 5%? Please check and/or correct

Reply: Many thanks: We corrected the value.

Line: 1083 Please check and correct font family

Reply: you are right. We corrected the font family

Line 1152: Fonts on maps are of a bad quality. Please increase resolution

Reply: We increase the Quality of Figure D1 and D2.

Round 2

Reviewer 1 Report

I am satisfied with the changes made by the authors.  I would encourage the authors to continue to shorten the manuscript prior to publication.